

# The Interactions between Soil-Biosphere-Atmosphere (ISBA) land surface model Multi-Energy Balance (MEB) option in SURFEX - Part 2: Model evaluation for local scale forest sites

Adrien Napoly[1], Aaron Boone[1], Patrick Samuelsson[2], Stefan Gollvik[2],
Eric Martin[3], Roland Seferian[1], Dominique Carrer[1], Bertrand Decharme[1], and
Lionel Jarlan[4]

[1]CNRM UMR 3589, Météo-France/CNRS, Toulouse, France
[2]Swedish Meteorological and Hydrological Institute, Norrköping, Sweden
[3]IRSTEA, U-R RECOVER, Aix en Provence, FRANCE
[4]Centre d'études Spatiales de la Biosphère (CESBIO), Toulouse, France

*Correspondence to:* Adrien Napoly (adr.napoly@gmail.com)

**Abstract.** Land surface models (LSMs) need to balance a complicated trade-off between computational cost and complexity in order to adequately represent the exchanges of energy, water and matter with the atmosphere and the ocean. Some current generation LSMs use a simplified or composite canopy approach that generates recurrent errors in simulated soil temperature and turbulent

fluxes. In response to these issues, a new version of the Interactions between the Surface Biosphere Atmosphere (ISBA) land surface model has recently been developed which explicitly solves the transfer of energy and water from the upper canopy and the forest floor which is characterized as a litter layer. The so-called Multi Energy Balance (MEB) version of ISBA is first evaluated for three well-instrumented contrasting local scale sites, and sensitivity tests are performed to explore the be-

havior of new model parameters. Second, ISBA-MEB is benchmarked against observations from 42 forested sites from the global micro-meteorological network (FluxNet) for multiple annual cycles.

It is shown that ISBA-MEB outperforms the composite version of ISBA in improving the representation of soil temperature, ground, sensible and to a lesser extent latent heat fluxes. Both versions of ISBA give comparable results in terms of simulated latent heat flux because of the similar formu-

lations of the water uptake and the stomatal resistance. However, MEB produces a better agreement with the observations of sensible heat flux than the previous version of ISBA for 87.5 % of the simulated years across the 42 forested FluxNet sites. Most of this improvement arises owing to the improved simulation of the ground conduction flux, which is greatly improved using MEB, especially owing to the forest litter parameterization. It is also shown that certain processes are also modeled

more realistically (such as the partitioning of evapotranspiration into transpiration and ground evap-



oration), even if certain statistical performances are neutral. The analyses demonstrate that shading effect of the vegetation, the explicit treatment of turbulent transfer for the canopy and ground, and the insulating thermal and hydrological effects of the forest floor litter turn out to be essential for simulating the exchange of energy, water and matter across a large range of forest types and climates.

## 1 Introduction

The land surface model (LSM) is one of the key parametrization schemes of atmospheric models used for numerical weather prediction and climate simulations. It is used to compute the turbulent fluxes of heat, water and momentum, along with the radiative fluxes at the surface-atmosphere interface. In addition, the current generation of LSMs are used to compute flux exchanges of certain chemical species (such as carbon dioxide and biogenic organic volatile carbon) and emissions of particles (such as aerosols from dust or biomass burning). The interaction soil biosphere atmosphere model (ISBA) is part of the SURFace EXternalisée platform (SURFEX) developed in recent years at Météo-France (Masson et al., 2013) and a suite of international partners. It is used to either a suit of surface schemes in either offline or coupled mode with an atmospheric model and/or a hydrological model. ISBA has benefited from continuous improvements since its first version (Noilhan and Planton, 1989), in particular for the carbon cycle (Calvet et al., 1998; Gibelin et al., 2006), soil mass and heat transfer (Boone et al., 1999; Decharme et al., 2011a), snowpack processes (Boone and Etchevers, 2001; Decharme et al., 2016), sub-grid hydrology (Decharme and Douville, 2006), and radiative transfer through the canopy (Carrer et al., 2013).

In order to remain consistent with the aforementioned developments and to respond to both current and future user demands, a multi energy balance (MEB) approach has been developed (Boone et al., 2016). This improvement consists in representing the surface as a so-called multi-source model (i.e. a separation of the canopy and the surface soil layer) in contrast to the standard ISBA soil-vegetation composite model. This new model has been designed to better represent certain processes for forested areas, and to incorporate new modeled processes which can benefit from a more accurate representation of the soil-vegetation continuum. In particular, partitioning of the incoming energy into turbulent and ground heat fluxes are expected to be more realistic with the explicit vegetation scheme as well as the partitioning of latent heat flux into its different components. The carbon cycle should also benefits from a more conceptually accurate representation of the surface (Carrer et al., 2013) since it is strongly linked to leaf temperature via assimilation of atmosphere Carbon. Moreover, snow melt during spring is very sensitive to the snow net radiation budget, which is impacted by the presence of tall vegetation, and interception and loss by the canopy during the winter season (Rutter et al., 2009).

The surface in forested regions beneath the canopy is often covered by a layer of dead leaves or needles, branches, fruits and other organic material which can be characterized as a litter layer. The



explicit inclusion of such a layer is neglected for the most part in LSMs used in global scale models, or it is only partly taken into account. For example, Sakaguchi and Zeng (2009) implemented a specific resistance due to litter for soil evaporation. The inclusion of litter-related processes has been shown to have a substantial impact on hydrological (Putuhena and Cordery, 1996; Guevara-Escobar et al., 2007) and thermal processes (Andrade et al., 2010). Litter has an important insulating capacity as its thermal diffusivity is small compared to soil (Riha et al., 1980), which leads to a strong impact on the soil temperature diurnal cycle and ground conduction flux. Changes in this flux can then impact the energy available for radiative and turbulent fluxes. The presence of litter also alters ground evaporation by covering the soil with a high porosity layer which essentially prevents liquid water capillary rise (Schaap and Bouten, 1997). It also constitutes another water interception reservoir for rainfall and canopy drip prior to infiltration and runoff (e.g., Gerrits et al., 2007). Some models have introduced parameterizations for litter (Ogée and Brunet, 2002; Enrique et al., 1999; Wilson et al., 2012; Haverd and Cuntz, 2010; Sakaguchi and Zeng, 2009), but the approach can be very different from one to another depending on their complexity (only modifying or adding a ground resistance, modeling the litter using an explicit single or multi-layer model). In addition, each of the models were validated against observations made for a single well-instrumented site.

The goal of the current study is to evaluate the impact of a new parametrization of the soil-litter-vegetation-atmosphere continuum at the local scale. In the first part of this study, an in-depth evaluation for three forest sites in France is carried out. They have been selected in order to represent a range of forest types and climates. The main goal is to understand the effect of both the explicit canopy layer and the litter layer on the available energy partitioning (latent, sensible, ground), soil temperatures and soil water content. The second part of this study consists in a benchmark study using 42 sites from the Fluxnet network (Baldocchi et al., 2001). The objective is twofold: (1) to evaluate the performance of the MEB options against the classic composite soil-vegetation version of the ISBA model over a wide range of forest species and climates; and (2) to analyze whether the general improvements seen at the three well-instrumented and documented local scale sites in France can be extended to other climates and forest types. This is essential since, as part of the SURFEX platform, ISBA is used in various configurations from kilometer resolutions at the regional scale such as the operational mesoscale numerical weather prediction model AROME, (Seity et al., 2011), or the operational distributed hydrological model system SIM, (Habets et al., 2008), and in the global climate models CNRM-CM5.1 (Voldoire et al., 2013) and CNRM-ESM1 (Séférian et al., 2015).

## 2 Model

### 2.1 The standard ISBA composite soil-vegetation model

The standard ISBA model uses a single composite soil-vegetation surface energy budget which means that the the properties of the soil and vegetation are aggregated within each grid-cell point



(Noilhan and Planton, 1989; Noilhan and Mahfouf, 1996). In this study, the so-called multi-layer soil diffusion (DIF) option is used, along with the explicit multi-layer snow scheme. The prognostic heat storage variables consist in the temperatures, $T_g$, for $N_g$ soil layers. The snowpack uses the snow enthalpy as a prognostic variable, from which both the snow liquid water content and temperature can be diagnosed for $N_n$ layers. The hydrological prognostic variables are the canopy reservoir for water interception, $W_r$ (kg m$^{-2}$), the snow liquid water content, $W_n$, for $N_n$ snow layers, and the soil volumetric water and liquid water equivalent ice contents, $w_g$ and $w_{gf}$ (m$^3$ m$^{-3}$), respectively, for $N_{g,l}$ active hydrological soil layers (where $N_{g,l} \leq N_g$). Further details describing the soil and snowpack representations as well as other prognostic variables and model assumptions are detailed in Decharme et al. (2016).

### 2.2 The ISBA-MEB model

The multi energy balance model, ISBA-MEB, treats up to three fully coupled distinct surface energy budgets (i.e. the snow surface, the bulk vegetation canopy and the ground which is characterized as either a soil surface or litter layer: see Section 2.3). The reader is referred to Boone et al. (2016) for an extended description of the various assumptions of the MEB approach, its full set of governing equations and its numerical aspects. Compared to the classic ISBA approach, there is one additional prognostic heat storage variable which is the vegetation temperature, $T_v$. The new hydrological prognostic variable is the liquid water equivalent snow stored within the vegetation canopy, $W_{rn}$ (kg m$^{-2}$).

### 2.3 Explicit model for litter in ISBA-MEB

Two methods are generally used to represent the effect of a litter layer within LSMs. The first method consists in adding a specific ground resistance in order to reduce the soil evaporation due to the presence of a litter layer. For example, Sakaguchi and Zeng (2009) and Park et al. (1998) have used this method in their models with the resistance depending on the leaf area index ($LAI$) of the litter or its thickness. The second method utilizes an explicit model to represent the effects of the litter. The implementation of this method turns out to be quite variable among different LSMs, although generally it accounts for both thermal and hydrological effects. In the current study, a model was introduced which models the first order effects of a litter layer on the surface energy and water budgets while minimizing the number of parameters and the complexity. To this end, a single bulk-layer approach, which is based on the model of (Schaap and Bouten, 1997),has been developed. The aforementioned study has also inspired similar approaches by (Ogée and Brunet, 2002) and (Haverd and Cuntz, 2010). The method is relatively simple: a single explicit litter layer is inserted between the vegetation layer and the upper soil layer. With this approach, three additional state equations have to be solved: i) the litter average temperature, $T_l$ (K), ii) the litter liquid water content, $W_l$ (kg m$^{-2}$), and iii) the litter liquid water equivalent ice content, $W_{lf}$ (kg m$^{-2}$). Two noteworthy assumptions





have been made: first, it is assumed that that there is no evaporation directly from the soil in the presence of litter, and second, water movement by capillarity rise from the soil upwards into litter is neglected. The energy budget is computed for the litter layer using prescribed values of thermal conductivity, heat capacity and litter thickness. The water budget is computed in a similar manner as

for the vegetation canopy layer. Finally, frozen water in the litter is modeled using the same simple freezing model method as for the soil in ISBA (Boone et al., 2000). The model is fully described in Appendix A.

## 3 Data

### 3.1 Energy balance closure

Energy balance closure is a well-known issue when turbulent fluxes are computed with the eddy covariance technique. The closure, $\delta$, is defined as the ratio of turbulent fluxes to available energy at the surface:

$$\delta = \frac{H + LE}{R_n - G - S} \tag{1}$$

where $S$, $R_n$, $G$, $H$ and $LE$ (W m$^{-2}$) represent the vegetation heat storage, the net radiation, ground

conduction (heat) flux, the sensible heat flux and the latent heat flux, respectively. The basic idea is that closure should be as close as possible to unity when observations are compared with LSM output since such models close the energy budget (to a high degree) by design. As it turns out, the mean closure imbalance for FluxNet sites is typically 20 % (Wilson et al., 2002). According to Twine et al. (2000), the net radiation is probably the most accurately measured component of the energy balance

and closure can be reasonably done by adjusting $H$ and $LE$. The Bowen ratio closure method is often adopted to correct the non-closure. It consists in assuming that the Bowen ratio (ratio of the sensible to the latent heat fluxes) is well estimated by the eddy covariance system, so that the turbulent fluxes can be adjusted while conserving the Bowen ratio to ensure closure (Blyth et al., 2010; Zheng et al., 2014; Er-Raki et al., 2008). The adjusted sensible and latent heat fluxes can be computed as:

$$H_{adj} = H + \text{res} \times \frac{H}{H + LE} \tag{2}$$

$$LE_{adj} = LE + \text{res} \times \frac{LE}{H + LE} \tag{3}$$

where $H_{adj}$ and $LE_{adj}$ (W m$^{-2}$) represent the adjusted sensible and latent heat fluxes, respectively, and the residual energy is defined as

$$\text{res} = R_n - G - S - H - LE \tag{4}$$

Among the terms in Eq. 4, the energy stored in the canopy, $S$, is generally not measured. Even though it is relatively small compared to the net radiation, it has been shown to be non-negligible



in some forest canopies (Oliphant et al., 2004). One possible method which can be used to address this issue is to use the simulated residual (Blyth et al., 2010). In the current study, tests were done using both the model simulated $S$ and neglecting it. It was found that incorporating this term had very little influence on the adjusted fluxes. In the analysis that follows, we neglected this term and both the original and the adjusted fluxes are shown.

### 3.2 Local scale evaluation over France

Three well-instrumented forest sites have been used for model evaluation which cover a range in climate, soils and vegetation characteristics. Their location is shown in Fig. 1 and their main characteristics are summarized in Table 2. At each site, measurements of sensible and latent heat fluxes were made using the eddy covariance technique, and the ground heat flux was measured with flux plates. Soil temperature ($T$) and volumetric moisture content ($w_g$) profiles were also available for two of the sites (Le Bray and Barbeau).

The Barbeau site is located in the Barbeau National Forest which is approximately 60 km from Paris (48.4°N, 2.7°E, France, elevation 90 m). This site consists mainly of a deciduous broadleaf mature oak forest which has an average height of 27 m. The climate is temperate, with a mean annual temperature of 10.7 °C and the annual rainfall is 680 kg m$^{-2}$. The soil texture is loam in the upper soil and clay loam at the lower portion of the soil profile (Prévost-Bouré et al., 2009).

The Le Bray site is located about 20 km from Bordeaux, France (44.7°N, -0.7°E, elevation 62 m). It is a maritime pine forest classified as evergreen needleleaf with an average height of 18 m with a dense grass understory which comprises about half of the total $LAI$ at its maximum. The mean annual temperature is 12.9 °C and the mean annual precipitation is 997 kg m$^{-2}$. The soil is a sandy and hydromorphic podzol with a layer of compacted sand which starts at a depth between 0.4 and 0.8 m, which constitutes the limit of root penetration. The site also has a water table whose depth fluctuates during the year and, at times, can reach the surface. This is the only site where specific measurements for litter were made, and a near constant litter thickness of 0.05 m was observed (Ogée and Brunet, 2002).

The Puéchabon site is located roughly 35 km northwest of Montpelier in the Puéchabon State Forest (43.7°N, 3.5°E, elevation 270 m). Vegetation is largely dominated by a dense over-story of holm oak (evergreen broadleaf forest) with a mean height of 5.5 m. The climate is typical Mediterranean with mean annual temperature of 13.5 °C and a mean annual precipitation of 872 kg m$^{-2}$. The soil texture is homogeneous down to about 0.5 m depth and it can be described as silty clay loam. The parent rock is limestone (Rambal et al., 2014)

### 3.3 Benchmark

The second part of the study uses observations from a subset of the FluxNet sites to asses systematically or benchmark the suite of ISBA versions developed at Météo-France. The FluxNet database



has been used for LSM evaluation by several widely used LSM models (Stöckli et al., 2008; Blyth et al., 2010; Ukkola et al., 2015). Many sites are available within the FluxNet database, however in
the current study, only those sites and years with an energy balance closure of 20 % (or less) before adjustment of the turbulent fluxes are retained. Ground heat flux was assumed to represent 3 % of net radiation when it is not available (this is close to the average value over the three French sites). After screening, 42 forested sites remain which give a total of 179 years of observations (see Fig. 11 for the locations of the sites used in the current study). The method used to gap fill missing atmospheric
forcing data is described in Vuichard and Papale (2015). The three sites from the first part of this study have been removed in this analysis.

## 4   Model set up

The simulations are performed using the diffusive soil (DIF) option meaning that the soil heat and mass transfers are solved on a multi-layer grid (Decharme et al., 2011b). A recently developed multi-
layer canopy radiative transfer scheme (Carrer et al., 2013) has been incorporated into MEB in order to determine the fraction of incoming radiation intercepted by the canopy layer, transmitted to the ground and reflected. This model uses structural organization of the canopy and spectral properties of the leaves as well as the albedo from the soil and the vegetation for two different spectral bands (visible and near-infrared). Finally, the estimation of the stomatal resistance is made with the A-
gs formulation (Calvet et al., 1998; Gibelin et al., 2006) which simulates photosynthesis and its coupling to the stomatal conductance in response to atmospheric $CO_2$. Parameters of the stomatal resistance scheme are vegetation type dependent and were chosen according to the ECOCLIMAP database.

Three simulations were performed for each site in order to assess the impact of the new canopy
and litter layers on the simulated fluxes, soil temperature and soil moisture. The models were forced with atmospheric data at half-hourly time steps. In the first simulation, the reference model (i.e. using a single composite vegetation and surface-soil layer) was used and it is hereafter referred to as ISBA. The second simulation with the explicit canopy layer corresponds to the ISBA model using the MEB approach (referred as MEB herein for simplicity). The last simulation was carried out with
both the explicit canopy layer and the explicit forest litter layer and it is referred to as MEBL.

### 4.1   Local scale

The pedotransfer functions of Clapp and Hornberger (1978) are used for the soil water flow. The soil hydraulic conductivity at saturation, the so-called $b$-exponent and the matric potential at saturation are computed with the continuous formulations of Noilhan et al. (1995) using the textural properties
at each site. The wilting point, field capacity and the water content at saturation were computed using the observations of soil volumetric water content and/or following the recommendations of



the site principal investigators (Table 3). An hydraulic conductivity exponential profile was used to model the packing of the soil with depth. Soil organic content, which affects the hydrological and thermal coefficients as in Decharme et al. (2016), is modeled using the input data from the Harmonized World Soil Database (HWSD, Nachtergaele and Batjes, 2012). $LAI$, canopy height, vegetation type, total soil and root depths were set to values found in the corresponding literature for each site (Table 3). Litter thickness, $\Delta z_l$, which is one of the key parameters of the litter module, varies in space and time. This study assumes it is constant in time based on estimates made at each site. This is an important approximation, but a literature review reveals that measured litter thickness generally varies from one to ten centimeters (see Table 1). Because of the uncertainties related to specification of this parameter, a series of sensitivity tests are done and the results are presented in Sec. 5.2.

At the Le Bray site, the water table has a significant influence on the seasonal soil wetness. Measurements of its depth were available and allowed a simple parametrization to be developed. It consists in a strong relaxation towards saturation in soil layers below the observed water table depth. Thus, soil moisture within the saturated zone is very close to the observed values in the saturated or nearly saturated layers, while soil moisture above this zone is permitted to freely evolve.

For each site, simulated turbulent heat fluxes were compared (Figs. 4a-b; 5a-b; 6a-b) to both the original and the adjusted-observed values assuming that model results should fall inside the area delimited by these two curves. Moreover, since a proper evaluation of each flux component of the energy balance has to be done with a closed energy budget (in order to be consistent with the model which imposes closure by design), the scores are computed with the adjusted flux values.

### 4.2 Benchmark

The ECOCLIMAP land cover and the HWSD soil data bases were used to prescribe most of the needed parameters as was done for the local scale evaluation for the three French sites (in the previous section). This is also consistent with the method used for spatially distributed offline and fully coupled (with the atmosphere) simulations with ISBA. However, note that soil texture, canopy height and vegetation type were chosen in agreement with literature values for each site where available, therefore they superseded the ECOCLIMAP values for these parameters. The default thickness of the litter layer is set to three centimeters based on the sensitivity test results presented in Section 5.2 and in overall agreement with literature values (Table 1).

Initial conditions can have a significant impact on both the sensible and the latent heat fluxes, but they were not known thus a spin-up period was used for each site. Sites were initialized with saturated soil water content conditions and the first available year was repeated at least ten times until a predefined convergence criteria was achieved. Only the latent and sensible heat fluxes are evaluated since the ground heat flux, soil temperature and soil water content were not available at each FluxNet site.





## 5 Results and discussion

The MEB model has been compared to both observations and the standard ISBA composite veg-
etation model for several well-instrumented contrasting (in terms of forest type and climate) sites
in France, and to a sub-set of the FluxNet sites for multiple annual cycles using a benchmarking
application. The results of these evaluations and of several sensitivity tests are given in this section.

### 5.1 Evaluation for three well-instrumented forested sites

#### 5.1.1 Net radiation

The total net radiation ($R_{net}$) is analyzed in terms of the net longwave ($LW_{net}$) and shortwave
($SW_{net}$) components. The $SW_{net}$ primarily depends on the prescribed albedo for the soil, $\alpha_g$,
and the vegetation, $\alpha_v$. In ISBA, the is net shortwave radiation is simply defined as $SW_{net} =
(1 - \alpha_{eff}) SW \downarrow$, where the effective albedo is simply $\alpha_{eff} = veg \, \alpha_v + (1 - veg) \alpha_g$ and $veg$ is the
vegetation cover fraction (which is constant for forests). For the MEB simulations, the scheme from
Carrer et al. (2013) is used which uses the soil and vegetation albedo for two spectral bands (which
are aggregated to all-wavelength values for ISBA), along with the $LAI$ (the notion of $veg$ is not
applicable to MEB) in order to obtain an estimate of the vegetation optical thickness. Despite these
differences, results are very close, as seen in Table 4. The models perform well with relatively low
values of average RMSE (6.1 W m$^{-2}$) and average annual error (AE < 8 W m$^{-2}$) for all three sites.
Since MEB explicitly uses the shortwave radiation transmitted through the canopy for the ground net
radiation computation, it can be compared with the photosynthetically active radiation ($PAR$) mea-
surements below the canopy within the Barbeau and Puéchabon forests (Fig. 3). Comparison with
MEB results revealed that the transmission coefficient for the $PAR$ is overestimated. The radiative
transfer scheme parameter which exerts the most control over this process is the clumping index,
and it has been adjusted for these two sites. However, the shortwave radiation transmission remains
slightly over-estimated at the Barbeau forest during the first three months of the year including this
adjustment (Fig. 3b). The main reason is that during this period, the $LAI$ is very low but the trunks
and branches intercept solar radiation (the forest is relatively old and tall, so that this effect can be
quite significant) which is not taken into account by the model. This could be done by including a
stem area index ($SAI$), but currently the default minimum $LAI$ is used as a proxy for this effect.

The simulated $LW_{net}$, which depends on the explicit contributions from the soil, vegetation and
snow in MEB, and the composite soil-vegetation layer and snow in ISBA, was quite similar among
the model versions and led to a fairly good comparison with measurements. The annual RMSE is less
than 8 W m$^{-2}$ for each site and run on average, and the annual AE is less than 3 W m$^{-2}$ (Table 4).
This is due, in part, to the use of the same values of emissivity for soil, snow and vegetation in
ISBA and MEB. It is also due to the fact that $LW \downarrow$ is the same for both models. Note that another
parameter which can have an impact on $LW_{net}$ is the ratio of the roughness length for momentum





to that for heat, which is commonly referred to as $kB^{-1} = ln(z_0/z_{0h})$. In the default version of ISBA, the roughness length ratio is defined as $z_0/z_{0h} = 10$ (which corresponds to $kB^{-1} \approx 2.3$),

and this value has been determined as a value which works well generally (although for local scale applications, values can be prescribed to range from 1 to on the order of $1 \times 10^2$). But note that there is evidence that $kB^{-1}$ is actually closer to 1 for tall canopies (e.g., Yang and Friedl, 2003) which corresponds to a roughness length ratio of approximately 2.7. A ratio of one is used for forests within the default version of the original two-source model in the RCA dual-energy budget LSM.

Thus currently, a ratio of one is used for MEB since it is closer to $kB^{-1} \approx 1$, in contrast to 10 for ISBA. The larger roughness ratio in ISBA can be seen as a way of compensating for an underestimated diurnal cycle of the surface radiative temperature owing to a composite soil-vegetation heat capacity (which is larger than that for a bulk vegetation layer as in MEB). This implies that the explicit canopy representation in MEB allows us to use a slightly more realistic value of roughness

length ratio for forest canopies.

### 5.1.2 Latent heat flux

At Le Bray and Puéchabon, the simulated values of total $LE$ are relatively close between the three simulations and in fairly good agreement with the adjusted measurements. Above the forest, most of the evapotranspiration comes from canopy transpiration (Fig. 2), which is modulated to a large

extent for the three sites by the root zone soil moisture. All three of the simulations use the same scheme for the soil water and the stomatal resistance. The simulations tend slightly underestimate $LE$ (negative AE) for both these sites and models on an annual basis. One possible explanation is that the formulation of the stomatal resistance was originally adapted for low vegetation types, such as crops (Calvet et al., 1998). At Barbeau, the $LE$ underestimation (bias) is larger. The likely reason

is owing to the presence of a water table that was not explicitly measured, but strong evidence for it's existence can be seen in the measured soil water (especially in winter and spring: Fig. 9e). Without an explicit groundwater parameterization, it is not possible to model this water table accurately and development of such a scheme was beyond the scope of the current study. But these very wet observed soil conditions resulted in mostly unstressed conditions during this time frame, and this

high level of soil water was not modeled by ISBA or MEB.

For the Mediterranean forest at Puéchabon, most of the net radiation is converted into sensible heat flux (Fig. 5a) which leads to good results for the three simulations (Table 4). However, the $R^2$ for $LE$ is still improved by 14 % for MEBL compared to ISBA (Table 4), which is related to the improved sensible heat flux simulation (see Section 5.1.3 for more details). The RMSE scores

are not very different between the three simulations, although the results are slightly improved with MEB and MEBL (the RMSE is 7 % lower with MEB and 10 % with MEBL compared to ISBA). The main difference between the three simulations for this site is the ground evaporation. It is very small for ISBA, but this is largely due to the class-dependent prescription of a vegetation fraction,



$veg$, of 0.99 (Table 3). This severely limits the baresoil evaporation for this particular class. This high
value was defined for this class within ISBA to avoid excessive baresoil evaporation in the composite
scheme (amounting to a tuning parameter) resulting from the use of an aggregated surface roughness
(resulting in a relatively large roughness for the ground) and to model in a very simplistic way the
shading effect. For MEB, the $veg$ parameter does not exist, and instead the partitioning between $E_g$
and $E_v$ is controlled more by physical processes (largely by the annual cycle of $LAI$). The MEB $E_g$
turns out to be significantly larger than that for ISBA for this site, but MEBL produces an $E_g$ which
is closer to that of ISBA.

The two other sites have a higher annual evapotranspiration than the Mediterranean site and a
larger ground evaporation, especially for ISBA. At Le Bray, evapotranspiration is almost the same for
the three simulations (Fig. 2a), with a slight underestimation between September and April (Fig. 4b).
Ground evaporation was measured at this site from March 14 to April 6 in 1998 Lamaud et al. (2001),
leading to an estimate of the cumulative evaporation from the ground of approximately 7.2 kg m$^{-2}$
during this period. For each of the three available years of this study (2006-2008), MEBL simulates
an evaporation from the litter of around 6.5 kg m$^{-2}$ during this period, whereas ISBA evaporates
16 kg m$^{-2}$ from the ground. This shows the ability of the litter model to limit ground evaporation
during this period.

At Barbeau, the interpretation of the results is much more complex because of the deciduous
broadleaf nature of the forest. In winter, the $LAI$ is very low and latent heat flux is dominated by
the ground evaporation. In spring, when net radiation increases but vegetation is not fully developed,
MEB significantly overestimates $LE$ owing to a large ground evaporation (Fig. 6b), in contrast to
MEBL, which limits surface evaporation to 1.5 times less owing to the litter layer. In summer, the
ISBA simulation appears to better simulate $LE$ (in terms of total amount). However, taking a closer
look at the partitioning of evapotranspiration reveals that transpiration and evaporation from the
canopy reservoir are almost the same but ground evaporation is very different compared to MEB
and MEBL. Both MEB simulations simulate considerably less ground evaporation compared to the
standard ISBA simulation which simulates 25 % of annual $LE$ as ground evaporation as shown in
Fig. 2 (especially due to the summertime contribution). This value is certainly over-estimated since at
this time of the year, very little energy is expected to reach the ground (the leaf area index is around
5 m$^2$ m$^{-2}$). For example, during July, ISBA and MEBL evaporate 33.4 kg m$^{-2}$ and 4.5 kg m$^{-2}$
directly from the ground, respectively, whereas the total energy reaching the soil, is only equivalent
to a maximum possible evaporation of 8.1 kg m$^{-2}$ based on $PAR$ measurements. Second, in the
composite ISBA scheme, the roughness length, $z_0$, used for ground evaporation is the same as that
for transpiration, and it is computed using an inverse-log averaging between soil and vegetation
values weighted by the fraction cover Noilhan et al. (1995). The forest cover in ISBA uses a fraction
cover of 0.95, thus the effective surface roughness is dominated by the vegetation roughness and
is nearly 2 orders of magnitude larger than a typical soil roughness (0.01 m, which is close to the





MEB value) leading to a relatively low aerodynamic resistance. Thus, the ISBA model simulates a reasonable $LE$ which is comparable to MEB owing to a weak transpiration compensating an excessive bare soil evaporation. This is one of the known biases which MEB was intended to reduce.

### 5.1.3 Sensible heat flux

For all of the sites, more significant differences occur for the simulated $H$ compared to $LE$; with the most significant differences occurring with MEBL (see Table 4). At Le Bray and Puéchabon, the observed $H$ is in better agreement between the MEBL and MEB runs with an average RMSE over these two sites of 67.7 W m$^{-2}$ with the ISBA, 49.5 W m$^{-2}$ with MEB and 48 W m$^{-2}$ for MEBL. In addition, the R$^2$ are improved with average of 0.82 for ISBA, 0.89 for MEB and 0.90 for MEBL. At

Barbeau during summer, the MEB and MEBL simulations constantly over-estimate $H$. As shown in the previous section, $LE$ is underestimated in particular for MEB and MEBL. As net radiation and ground heat flux (section 5.1.4) are well simulated, $H$ tends to be over-estimated. ISBA produces fairly good results despite overestimations of the ground heat flux and ground evaporation.

The improvement for the two MEB simulations are mainly related to three processes;

– The use of an explicit canopy layer which intercepts most of the downward solar radiation thereby reducing the net radiation at the ground surface. This leaves more energy available for turbulent fluxes in contrast to ISBA which is directly connected to forest floor and can more easily propagate energy into the ground by conduction.

– The use of a lower roughness length ratio (momentum to heat: $z_0/z_{0h}$) which leads to an
increase in the roughness length for heat and water vapor and therefore the turbulent fluxes.

– The presence of a litter layer limits the penetration of energy into the ground because of its low thermal diffusivity which leads to a reduction of the ground heat flux (amplitude) and thus to more available energy for $H$ and $LE$.

### 5.1.4 Ground heat flux

A substantial effect of both MEB and MEBL is the ground heat flux reduction. The explicit representation of the canopy induces a shading effect due to the leaves, stems and branches, so that less energy reaches the ground. But in addition, the explicit representation of the litter layer also modifies the ground heat flux by acting as a buffer for the top soil layer due to its low thermal diffusivity. It can be seen in Fig. 7 that for each site, MEB and MEBL had improved simulations of $G$ in terms

of amplitude and phase with regards to ISBA. The mean AE of the three experiments remains very low (Table 4), even for the ISBA model. This demonstrates that the two MEB experiments reduce the amplitude and improve the phasing of this flux while its daily average was reasonably well represented with the standard ISBA model. The mean RMSE averaged over all years and sites is 11.7





W m$^{-2}$ with the explicit litter, 24.3 W m$^{-2}$ without it and 47.1 W m$^{-2}$ using ISBA (the scores are
summarized in Table 4).

The ISBA overestimation of the flux amplitude often represents several 10's of W m$^{-2}$ during
both daytime and nighttime (Fig.s 4c; 5c; 6c), which reduces the amount of energy available for
turbulent fluxes. For all sites, improvements of the ground heat flux are the same from one season
to another which means that using a constant thickness for the litter layer is likely adequate. The
MEBL overestimates the reduction of $G$ from June to September only for the Le Bray site (Fig. 4c).
However, the presence of a very dense understory at this site (accounting for half of the $LAI$ over
this period) makes the radiative transfer modeling much more complicated and could potentially
lead to an underestimation of the incoming radiation at the forest floor. The understory contribution
to the shortwave fluxes is simply represented using a bulk overstory-understory aggregated $LAI$.
In contrast to the other two sites, the effect of shading and litter at Puéchabon are not sufficient to
reduce the $G$ significantly (to be more in line with observed values) no matter what the season. The
parameters of the radiative transfer scheme have been calibrated through the modification of the
clumping index so that the energy passing through the canopy is well modeled (Fig 3b). Sensitivity
tests have been done for which the ground albedo and soil texture were modified over realistic ranges,
but this did not resolve the issue, and so the over-estimated $G$ is likely due to other causes (such as
the need to better adjust the turbulent transfer parameters to better correspond to the vegetation at
this site). At any rate, the MEBL $G$ is still superior to that simulated using MEB and ISBA for this
site. Finally, at Barbeau, the MEBL simulation produces good results for $G$ (Fig. 6c) except for a
short period in winter when the shortwave radiation transmission is also over-estimated (see Fig. 3c).

**5.1.5   Soil temperature**

A good description of the soil thermal characteristics is needed to model temperatures at different
depths, along with a correspondingly good estimate of the surface soil heat flux. The soil characteris-
tics (thermal conductivity and heat capacity) are calculated based on the input soil texture (sand and
clay fractions) and organic matter contained in the soil. The soil temperature simulation statistics
in Table 5 were computed for the three model configurations using available measurements at each
site. MEBL and MEB have considerably lower RMSE of soil temperatures for each site compared
to ISBA. The RMSE of the first measured depth of each site (between 0.04 and 0.10 m) is 10 to 20
% lower for MEB compared to the ISBA model, and 20 to 60 % lower between MEBL and ISBA.
Moreover, R$^2$ is also improved with an average value for all of the sites of 0.96 for MEBL, 0.94
with MEB and 0.90 for ISBA. The good agreement between simulated and measured temperatures
and the improvements with MEB are consistent with the improvement of the ground heat flux shown
in the previous section (Fig. 7). Note, however, that the improvement of the simulated temperature
at Puéchabon is not as significant as for the other sites. This was expected since, as shown in the
previous section, the $G$ was over-estimated for this site: the RMSE is reduced with the MEB and





MEBL experiments, but it remains high (3.8 and 3.3 K, respectively). In general, there are relatively
low AE values for each site and soil depth (AE < 1.3 K), but a seasonally varying AE remains. The
general effect of MEB is to reduce the mean annual temperature by 1 to 2 K (resulting in a shift in
the annual cycle), while MEBL dampens the annual cycle amplitude. Indeed, the litter reduces both
the energy loss during the cold season and gains in heat during the summer season. The composite

monthly average diurnal cycles of the uppermost available measured temperature for each site is
shown in Fig. 8. The relatively low AE calculated over the entire year (Table 5) masks a negative
AE during the cold season (November to March) and a positive AE during the warm season (May
to September). At all sites, both MEB simulations improve the mean daily values during the warm
season, but only MEBL reduces the negative bias that occurs during the cold season. So, when look-

ing at the composite diurnal cycles, it is obvious that the use of MEBL verses MEB improves the
performance in terms of amplitude and phase all year long even if a relatively small bias remains.
The ISBA model daytime overestimation can be as high as 10 K , whereas with the MEBL model it
does not exceed 3 K.

### 5.1.6  Soil moisture

The observed and simulated soil water content time series for an annual cycle are displayed in Fig. 9.
Soil volumetric water content measures are available at 30 minute time steps at Le Bray and Barbeau.
At Puéchabon, no half-hourly measurements were available but a reference curve, which corresponds
to a model-derived interpolation of discrete soil water storage measurement (Lempereur et al., 2015),
has been added to the figure. A fairly good overall agreement between simulated and observed total

soil water content is found. Moreover, the three experiments lead to very similar results, which is
consistent with the fact that evapotranspiration has been found to be the least impacted flux between
ISBA, MEB and MEBL.

More significant differences can be seen in terms of the top soil water content. In particular, for
the Le Bray and Barbeau forests, at the end and beginning of the year, respectively, ISBA and MEB

simulate unrealistic drops in the liquid soil water content (Fig. 9, b and f). These drops are caused
by the underestimation of the soil temperature during these periods which makes the water freeze.
These drops are not present in the MEBL simulation. In fact, the near-surface soil temperature stays
above 0 deg C owing to an insulating effect from the litter so that the water remains liquid as seen in
the measurements. At Barbeau, all the three model approaches underestimate the total soil moisture

during the first two thirds of the year (Fig. 9e) which is mainly related to the presence of a water
table which is not explicitly modeled. MEB and MEBL are more sensitive than ISBA to this process
since they simulate more transpiration than ISBA (although MEBL is slightly wetter since $E_g$ is
always less in MEBL compared to MEB.





### 5.2 Sensitivity tests

There is uncertainty with respect to the definition of several key model parameters which are not usually available in the observations so that several sensitivity tests have been undertaken. Three parameters have been tested; i) the extinction coefficient for the long wave transmission through the canopy (Boone et al., 2016), ii) the leaf geometry which modulates the solar radiation transmitted through the canopy (Carrer et al., 2013), and iii) the litter thickness which controls the impact of

litter on both the hydrological and thermal regimes (see Appendix A).

   For all three of the sites, the long wave transmission coefficient has a very weak influence on each of the simulations for a reasonable range of values (0.3 to 0.7, 0.5 being the default value which is based on an value used by Verseghy et al. (1993). The RMSE calculated with observed and simulated total long wave up radiation varies less than 2 W m$^{-2}$ over this range for the 3 sites on an annual

basis.

   The litter layer thickness values tested range between 0.01 and 0.10 m based on values from the literature (Table 1). The MEBL default values based on recommendations from the site principal investigators are shown in Table 2. The $G$ is the most sensitive to these changes and RMSE values for these tests are shown on Fig. 10 for Le Bray and Barbeau. Note that these tests were not done

for Puéchabon since $G$ was overestimated despite the litter layer (which implies that optimizing litter thickness for these site to improve fluxes would likely result as a compensating error related to another process). Minimum errors values are reached for thicknesses above approximately 0.03 m which is consistent with the recommended site values and those from several other studies (see Table 1). Note that these thickness values also produce the lowest $H$ and $LE$ RMSE (not shown).

The clumping index was tested since it is the key parameter of the radiative transfer controlling the transmission and absorption of incoming short wave radiation through the canopy (Carrer et al., 2013). Three configurations were tested for the MEBL configuration; i) the default value, ii) a clumping index increased by 50 % (a more closed canopy) and iii) an index decreased by 50 %. This parameter alters the computation of the upwelling shortwave radiation, and the values of the turbu-

lent and ground heat flux since it changes the partitioning of available energy between the canopy and the litter layer. However, the $H$ and $LE$ RMSE only changed by 3 to 8 % depending on site, thus for now, the default clumping index values are used.

   In summary, owing to these sensitivity tests, the default extinction coefficient for long wave transmission is retained as a constant value of 0.5. A default constant value of litter thickness is defined as

0.03 m since it is both not widely observed and it tends to be a threshold for which the effect of litter on $G$ becomes notable and positive (and errors begin to slowly increase slightly above this threshold) for several contrasting sites. One could even define an slightly larger value based on Fig. 10 (0.04 m for example), but a first order approximation is required as opposed to a tuned value based on just two sites. Also, the layer should be relatively thin to insure a robust diurnal cycle (as opposed to

using values on the high side from these tests or based on the literature). The default clumping index



used in the shortwave radiative transfer led to an over-estimation of the below-canopy radiation for the two sites where measurements of below canopy $PAR$ were available, and slight improvements were found by adjusting this parameter (to values still within it's realistic range). But in fact, the impact of tuning the clumping index was found to have only a relatively small impact on the simulated

turbulent heat fluxes with the MEBL simulation so that the default value is considered to be robust enough for now. As a final note, if measured values are available or can be readily determined for the longwave transmission coefficient, litter thickness, and shortwave radiation transmission (for example, at local scale well-instrumented sites for studies oriented towards detailed process analysis), they can be used to replace the aforementioned default values.

### 5.3   Worldwide skill score assessment


The comparison of modeling results with field measurements from over 42 Fluxnet sites (Fig. 11) reveals that the results from the benchmark are consistent with those from the three local French sites. Only the MEBL and the ISBA models are considered here.

MEBL and ISBA generally performed well in simulating the $R_{net}$, but both tended to slightly

underestimate daytime values with a mean bias of 10 W m$^{-2}$ corresponding to approximately 9 % of the average net radiation Note that the $R_{net}$ was slightly better simulated for the three local scale sites (Sec. 5.1), but this results in part, because the benchmark uses values from the ECOCLIMAP database (which likely introduces some error relative to using observed values of albedo for example). Scatter plot of the statistical metrics (RMSE, R$^2$ and AE) obtained for the sensible and the

latent heat fluxes are shown in Fig. 12 and Fig. 13, respectively. Each black dot corresponds to the average score over all available years for a given site. The shaded area corresponds to the sites and years for which the MEBL model outperforms the ISBA model and the dashed lines represent the average over all sites. Finally, the percentage of sites for which MEBL was superior to ISBA for a particular score is given in the title of each panel within the aforementioned figures.

The sensible heat flux AE (Fig. 12c) is quite similar for the two model approaches as evidenced by the relatively low scatter (most of the values are within -20 to 20 W m$^{-2}$). This is consistent with previous conclusions for the detailed local scale run analysis (Table 4) and confirms that the influence of the multi-source model is more significant for diurnal cycles than for daily average results. However, a very significant improvement of the sensible heat flux R$^2$ is observed and for

93.5 % of the sites for MEBL. This shows the ability of MEBL, as demonstrated in Sec. 5.1, to better represent the phase of turbulent fluxes owing to the explicit representation of the canopy layer and especially the more realistic lower value of the heat capacity of vegetation with respect to ISBA.

The amplitude of the sensible heat flux diurnal cycle is also improved as suggested by the improved RMSE values (Fig. 12a) over 90.5 % of the sites. This better amplitude performance is re-

lated to both the canopy shading and litter insulation effects which lower the diurnal amplitude of the ground heat flux and provide more energy for the turbulent fluxes. The recent so called PLUMBER





experiment Best et al. (2015) found that for $H$, most of the LSMs they studied (including ISBA) were not able to beat the one variable non-linear regression that uses instantaneous downward short-wave radiation and observed surface fluxes. The results in this study present a possible reason for this (since most of the LSMs do not include an explicit litter layer), therefore we recommend that the ground heat flux should be better investigated and could be contributing to the generally poor performance of simulating $H$ among LSMs for the Fluxnet sites.

The latent heat flux, $LE$, results between the two models are more similar than for $H$. Based on the results from Sec. 5.1, this was expected since the vapor flux is mainly controlled by the stomatal resistance parametrization and soil hydrology (which are the same for MEBL and ISBA). Fig. 13a shows that the differences in RMSE between MEBL and ISBA are relatively small between the two simulations. However, an improvement in the $R^2$ coefficient is obtained (Fig. 13b), with 73.3 % of the sites showing an improvement with MEBL. Note that a negative AE is obtained for most of the sites. This result is also consistent with the local evaluation over the three French sites (Table 4 and as shown in Fig. 2). Since both the ISBA and MEBL approaches have similar errors in this respect, it is assumed that this is likely caused by an aspect of ISBA which is common among the two models, such as root-zone water uptake, hydrology, or stomatal resistance for example. In a general, a very good consistency is found between the analysis in Sec. 5.1 for highly instrumented sites and the more general benchmark application in this section. This is a positive result for a model which is designed for weather forecast and climate simulations at a global scale.

Finally, it is of interest to determine if the results are conditioned by certain key physiographic parameters, notably the $LAI$. The differences in RMSE between MEBL and ISBA are shown in Fig. 14 for different ranges of $LAI$ as box plots. Each box is plotted using the 10, 25, 50, 75 and 90 percentiles of the normalized difference in RMSE calculated over one-month periods for each site that satisfies the closure condition adopted for this study (i.e. less than 20 % imbalance). In Fig. 14a , the improvement of RMSE is significant for forested sites with a relatively low $LAI$ ($LAI<2$) which represents 19 % of the cases, moderate for medium $LAI$ ($2<LAI<4$), 46 % of the cases, and weak for high $LAI$ ($LAI>4$), 35 % of the cases. This shows that the gains of the multi source energy budget are more significant when the canopy is relatively sparse or open (corresponding to a relatively low $LAI$), so that the surface fluxes have significant contributions from both the ground and the canopy. When there is a medium $LAI$ range, the same conclusion applies to a lesser extent. Thus as $LAI$ becomes large, the MEBL and ISBA results converge (since the fluxes are dominated by the canopy). Fig. 14 is consistent with previous results in that the simulated $LE$ is similar between MEBL and ISBA so that RMSE differences are low on average.



## 6  Conclusions

This study is the second of a set of two papers which describe the introduction of the new multi energy balance (MEB) approach within the interactions between the soil biosphere atmosphere model (ISBA) as part of the SURFEX platform. Two new explicit bulk layers have been implemented, one for the vegetation canopy and the other for a litter layer. This paper describes a two-part local-scale offline evaluation of both the bulk canopy scheme (MEB) and the combined bulk canopy-litter layer (MEBL) approaches, and the results are also compared to the standard composite-vegetation ISBA model. The model parameterization governing the litter layer is also presented. The evaluation is done by investigating the ability of the models to simulate the fluxes above (sensible and latent heat flux) and below (ground heat flux) different forest canopies and the ability of the different approaches to reproduce the observed soil temperatures and soil water content.

In the first part of this study, an evaluation over three well-instrumented forested sites in France was done using observed forest characteristics, and turbulent, radiative and heat conduction (ground flux) measurements with a particular attention paid to the energy balance closure issue. The mid-latitude forest sites were contrasting in terms of both vegetation type (needleleaf, broadleaf) and climate (Mediterranean, temperate). In the second part of this study, a statistical evaluation was done using the framework of a benchmark platform based upon 42 sites scattered throughout the world from the Fluxnet network.

In terms of the model evaluation for the three French forested sites, the standard ISBA model was found to underestimate the amplitude of the sensible heat flux, $H$, and overestimate that of the ground heat flux, $G$, the latter of which is in agreement with an overestimation of the soil temperature amplitude. Also, latent heat flux is generally well simulated despite being slightly underestimated. The simulation with the two new explicit layers greatly reduced the $G$ and soil temperature errors owing mainly to shading effect of the canopy layer and the low thermal diffusivity of the litter layer. The relatively low thermal conductivity of the litter greatly reduced the ground heat flux and soil temperature amplitudes, in addition to modifying their phase in better agreement with the measurements: the RMSE decreased from 47.1 to 10.0 W m$^{-2}$ on average. As a result, more energy was available for the turbulent heat fluxes. Since latent heat flux was generally water limited, most of this excess energy went into $H$: the average RMSE for the three well-instrumented sites decreased from 62.1 to 50 W m$^{-2}$ with the new parameterizations. This result might have importance beyond those for ISBA, since the results of the recent PLUMBER (Best et al., 2015) LSM inter-comparison and evaluation experiment highlighted the general poor performance of LSM-estimated $H$ (including ISBA) compared to a subset of the Fluxnet observations. Since net radiation and latent heat fluxes were generally reasonably well simulated, this implies that one of the main sources of error in $H$ is likely related to the ground heat flux simulation based on the energy budget equation. This study showed that at least for ISBA (using the MEBL option), the inclusion of litter lead to significantly improved ground conduction which resulted in better sensible heat fluxes.



In terms of temporal dynamics, the main differences in latent heat flux between ISBA and MEBL occur during spring for the deciduous forest site where the litter layer acts to significantly limit soil evaporation, whereas ISBA and MEB (without explicit litter) overestimate evapotranspiration due to

strong ground evaporation (owing to a relatively low $LAI$ combined with large incoming radiation and generally low to unstressed soil conditions). And despite the overall similar total annual evapotranspiration simulated by ISBA and MEBL, the partitioning between canopy evapotranspiration, $E_v$, and ground evaporation, $E_g$, is more realistic with MEBL than ISBA. This is mainly related to the conceptual design of the composite vegetation, which uses a single aggregated roughness length

for both vegetation and soil (which leads to a slight underestimate of canopy roughness but a significant overestimate of ground roughness and thus relatively large $E_g$ compared to the explicit bulk canopy approach of MEB). In addition, the structure of MEB enables the model to directly use the radiation transmitted through the canopy when computing the ground surface energy budget. Measurements from two of the three well-instrumented forest sites indicate that MEBL simulates the

downwelling shortwave radiation relatively well, and this sets a theoretical limit on the amount of energy available for the turbulent fluxes at the surface (which again, is often lower than the energy available for $E_g$ in the composite ISBA scheme). The more physically-based vegetation canopy in MEB also implies that there is no longer a dependence on the ISBA parameter $veg$ (vegetation cover fraction), which was constant for forests and tuned (to values between 0.95 and 0.99) to prevent

excessive $E_g$.

The main conclusions made from analysis of the three well-instrumented sites were found to be consistent with the results of a statistical benchmark analysis over a subset of 42 forested sites from the Fluxnet network. The sensible heat flux RMSE was improved for 87.5 % of these sites with the new parameterizations (MEBL). The selected sites encompassed a wide range of climate and

several different forest land-cover types, thus it is a necessary test before implementing MEBL in regional to global scale applications. The benchmark also showed that the impact of the explicit treatment of the canopy and litter layer was more significant for relatively open canopies (low to medium $LAI$ values), whereas for closed canopies (high $LAI$), all three of the model approaches simulation results converge. This is not overly surprising since in the limit as a canopy becomes tall

and quite dense, the composite scheme resembles a vegetation canopy (the soil contribution becomes significantly less). But again, ISBA tends to simulate considerably more baresoil evaporation in the peak growing season (maximum $LAI$) than MEBL, so there is error compensation which is masked to a large extent when looking at the total evapotranspiration. In terms of prospectives, evaluation of MEB is ongoing, and offline spatially distributed applications within SIM (Habets et al., 2008)

and at the global scale are being tested and compared to ISBA, along with an examination of the improvements obtained in high latitudes (owing, in part, to a more detailed representation of snow interception processes). Also, additional local scale tests are being done for both tropical (since there were relatively few such sites in Fluxnet) and semi-arid forests (owing to the impact of MEB for the



latter as shown herein). Finally, experiments are being prepared for fully coupled land-atmosphere

simulation tests in both mesoscale and climate model applications at Météo-France.

## 7   Code Availability

The MEB code is a part of the ISBA LSM and is available as open source via the surface modelling patform called SURFEX, which can be downloaded at http://www.cnrm-game-meteo.fr/surfex/. SUR-FEX is updated at a relatively low frequency (every 3 to 6 months) and the developments presented

in this paper are available starting with SURFEX version 8.0. If more frequent updates are needed, or if what is required is not in Open-SURFEX (DrHOOK, FA/LFI formats, GAUSSIAN grid), you are invited to follow the procedure to get a SVN account and to access real-time modifications of the code (see the instructions at the previous link).

*Acknowledgements.*  This work is a contribution to the ongoing efforts to improve the SURFEX platform. The

authors would like to thank the teams of the three French eddy flux towers involved in this study: Le Bray, Barbeau and Puéchabon, for making their data available, and to all of the contributors to the Fluxnet data base. This study was fully funded by a Météo-France grant.





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

**Appendix A:  Description of the litter layer model**

Forest litter is represented using a single model layer which generally ranges in thickness from 0.01 to 0.10 m, and in the absence of ancillary data, the default value is 0.03 m. When this option is active, an additional layer is added to the soil for the thermal and energy budget computations with litter-specific thermal properties. This means that the numerical solution method is identical to that presented in the companion paper by Boone et al. (2016). In terms of hydrology, an additional reservoir is added which uses a relatively simple bucket-type scheme with a litter-specific maximum water storage capacity. The model physics and governing equations are reviewed herein.

**A1  Prognostic equations**

The energy budget for the snow-free litter layer can be expressed as:

$$\mathcal{C}_l \frac{\partial T_l}{\partial t} = R_{nl} - H_l - LE_l - G_{g,1} + L_f \Phi_l \tag{A1}$$

where $T_l$ is the litter temperature (K), $\Delta z_l$ (m) is the thickness of the litter layer, and $\mathcal{C}_l$ (J K$^{-1}$ m$^{-2}$) is the effective heat capacity of the litter. $R_{n,l}$, $H_l$, $LE_l$, $G_{g,1}$ represent the net radiation, sensible heat flux, latent heat flux and ground conduction flux from the litter layer, respectively. Note that when litter is present, $R_{n,l}$, $H_l$, $LE_l$ correspond to the ground fluxes in Boone et al. (2016). The additional terms added owing to a snow cover are described in Boone et al. (2016), and are therefore not repeated here.

The liquid water content of the litter layer evolves following:

$$\frac{\partial W_l}{\partial t} = P_l - E_l - D_l - \Phi_l \qquad (0 < W_l < W_{l,max}) \tag{A2}$$

where $P_l$ is the sum of the rates of the rainfall passing through the canopy, canopy drip and ground-based snow melt, $E_l$ represents the litter evaporation rate, $D_l$ is the drainage rate from the litter to the soil (all in kg m$^{-2}$ s$^{-1}$). The maximum liquid water content in the litter reservoir is defined as $W_{l,max} = w_{l,max} \Delta z_l \rho_w$ (kg m$^{-2}$). The default value for the maximum holding capacity of the litter layer, $w_{l,max}$, is 0.12 m$^3$ m$^{-3}$ (Putuhena and Cordery, 1996). All water in excess of this maximum value is then partitioned between infiltration and surface runoff by the ISBA soil hydrological model.

The liquid water equivalent of ice contained in the litter layer is governed by:

$$\frac{\partial W_{lf}}{\partial t} = \Phi_l - E_{lf} \tag{A3}$$

where $E_{lf}$ represents the sublimation of ice contained within the litter layer.



### A2 Phase Change

The phase change rate, $\Phi_l$ (kg m$^{-2}$ s$^{-1}$), is defined as:

$$\Phi_l = \frac{1}{\tau_{ice}} \left\{ \delta_f \min \left[ \frac{\rho_i C_i \Delta z_l (T_l - T_f)}{L_f}, W_{lf} \right] + (1 - \delta_f) \min \left[ \frac{\rho_i C_i \Delta z_l (T_f - T_l)}{L_f}, W_l \right] \right\} \quad \text{(A4)}$$

where $L_f$ represents the latent heat of fusion (J kg$^{-1}$), $\rho_i$ is the density of ice (here defined as 920 kg m$^{-3}$), the freezing point temperature is $T_f = 273.15$ K, and $C_i$ is the specific heat capacity of ice ($2.106 \times 10^3$ J K$^{-1}$ kg$^{-1}$). The delta function $\delta_f = 1$ if energy is available for melting (i.e. $T_l - T_f > 0$), otherwise it is $\delta_f = 0$. $\tau_{ice}$ is a parameter which represents the characteristic time scale for phase changes (Giard and Bazile, 2000). The updated temperature is first computed from Eq. A1 with $\Phi_l = 0$, then the phase change is computed as an adjustment to $T_l$, $W_l$ and $W_{lf}$ as in Boone et al. (2000).

### A3 Energy Fluxes

It is assumed that litter below the canopy is spatially homogeneous so that it intercepts all of the incoming radiation. Thus, the net radiation $R_{nl}$ for the litter layer is the same that for the first soil layer in the standard model:

$$R_{n,g} = SW_{net\,g} + LW_{net\,g} \quad \text{(A5)}$$

where $SW_{net\,g}$ and $LW_{net\,g}$ correspond to the net shortwave and longwave radiation (W m$^{-2}$) as in Boone et al. (2016). Note that currently, the soil emissivity and albedo values are used for the litter for spatially distributed simulations pending the development of global datasets of these parameters for litter or the development of an appropriate model to estimate them. For local scale simulations, the values can be defined based on observations.

The below-canopy sensible heat flux, $H_l$ (W m$^{-2}$), is computed the same way that for the top soil layer in the ISBA model as:

$$H_l = \rho_a \frac{(\mathcal{T}_l - \mathcal{T}_c)}{R_{ag-c}} \quad \text{(A6)}$$

where $R_{ag-c}$ is the aerodynamic resistance between the ground and the canopy air space which is based on Choudhury and Monteith (1988). $\mathcal{T}_l$ and $\mathcal{T}_c$ (J kg$^{-1}$) are thermodynamic variables which are linearly related to temperature (Boone et al., 2016). The latent heat flux is partitioned between evaporation and sublimation in the litter layer:

$$LE_l = (1 - p_{lf}) LE_l + p_{lf} LE_{lf} \quad \text{(A7)}$$

where $p_{lf}$ is the fraction of frozen water in the litter layer and

$$LE_l = L_v \rho_a \frac{[h_{ul} q_{sat}(T_l) - q_c]}{R_{ag-c}} \quad \text{(A8a)}$$

$$LE_{lf} = L_s \rho_a \frac{[h_{ulf} q_{sat}(T_l) - q_c]}{R_{ag-c}} \quad \text{(A8b)}$$





where the specific humidity of the canopy air space is represented by $q_c$. The specific humidity at saturation over liquid water is represented by $q_{sat}$ (kg kg$^{-1}$). Note, it would be more accurate to use the specific humidity at saturation over ice in Eq. A8b, but this complicates the linearization and this effect is neglected for now (Boone et al., 2016). The surface humidity factors for liquid and frozen water are represented by $h_{ul}$ and $h_{ulf}$, respectively. They are computed as the relative humidity following (Noilhan and Planton, 1989):

$$h_{ul} = \frac{1}{2}\left[1 - \cos\left(\pi \frac{W_l}{W_{l,max}}\right)\right] \tag{A9}$$

Note that $h_{ulf}$ is computed by replacing $W_l$ and $W_{l,max}$ by the values for the liquid water equivalent ice content. The maximum liquid holding capacity is modified for ice following (Boone et al., 2000).

Finally, the ground conduction flux (W m$^{-2}$) between the litter layer and the underlying soil is computed as:

$$G_{g,1} = \frac{T_l - T_{g,1}}{(\Delta z_l / \lambda_l) + (\Delta z_{g,1}/\lambda_{g,1})} \tag{A10}$$

where $\lambda_l$ and $\lambda_{g,1}$ are the litter and first soil layer thermal conductivities respectively and $\Delta z_{g,1}$ is the thickness of the first soil layer.

**A4  Water interception and fluxes**

The water intercepted by the litter layer corresponds to the sum of the rain passing through the canopy and the drip from the canopy.

$$P_l = (1 - \sigma_v)P_r + D_c \tag{A11}$$

Note that when snow is present, melt from the snowpack is also included in this term (Boone et al., 2016). The fraction of the total rainfall $P_r$ (kg m$^{-3}$ s$^{-1}$) intercepted by the canopy is modulated by $\sigma_v$, which depends on the Leaf Area Index of the canopy. Finally, the canopy drip is represented by $D_c$. Further details on canopy interception and drip are given in Boone et al. (2016). The drainage from the litter is simply computed as in Noilhan and Mahfouf (1996):

$$D_l = \max(0, W_l - W_{l,max}) \tag{A12}$$

and the amount of $D_l$ which can infiltrate into the soil is limited by Darcy's law, with any residual contributing to surface runoff. Note that for simplicity, a gravitational drainage type formulation is not used for litter, but rather a tipping bucket following Ogée and Brunet (2002).

**A5  Thermal properties**

The litter thermal conductivity, $\lambda_l$ (W m$^{-1}$ K$^{-1}$), is computed according to De Vries (1963) as:

$$\lambda_l = 0.1 + 0.03\left(\frac{W_l}{\rho_w \Delta z_l}\right) \tag{A13}$$





The effective heat capacity of the litter, $\mathcal{C}_l$ (J m$^{-2}$ K$^{-1}$), is computed using

$$\mathcal{C}_l = \Delta z_l \rho_{ld} C_{ld} + W_l C_w + W_{lf} C_i \tag{A14}$$

where the specific heat capacity of liquid water is $C_w = 4.218 \times 10^3$ (J K$^{-1}$ kg$^{-1}$). The dry density of the litter is defined as $\rho_{ld}$. Ogée and Brunet (2002) used a value of dry litter density of 45 kg m$^{-3}$ for a pine forest. Meekins and McCarthy (2001) measured a litter density of 46 kg m$^{-3}$ in a

925 deciduous forest and Kostel-Hughes et al. (1998) estimated values varying between 27 to 38 kg m$^{-3}$ for oak forests. Currently ECOCLIMAP doesn't distinguish between different types of deciduous trees, thus by default, $\rho_{ld}$ is assigned a value of 45 kg m$^{-3}$. As a proxy for the specific heat of litter, we use the specific heat capacity of organic material from Farouki (1986) which is $C_{ld} = 1.926 \times 10^3$ J kg$^{-1}$ K$^{-1}$. Currently, constant values for $\rho_{ld}$ and $C_{ld}$ are used for spatially distributed applications

or on the local scale, unless observational data are available.

**Appendix B: Statistical Scores**

This three scores are defined respectively as:

$$R = \frac{\overline{(x_m - \overline{x_m})(x_o - \overline{x_o})}}{\sigma_m \sigma_o} \qquad SDV = \frac{\sigma_m}{\sigma_o} \qquad cRMSE = \frac{\sqrt{\overline{[(x_m - \overline{x_m}) - (x_o - \overline{x_o})]^2}}}{\sigma_m} \tag{B1}$$

where the overbar represents the average, $x_m$ and $x_o$ are the modeled and observed datasets respec-

935 tively and $\sigma$ is the standard deviation defined as:

$$\sigma = \sqrt{\overline{(x - \overline{x})^2}} \tag{B2}$$

The cRMSE difference between MEBL and ISBA is defined as:

$$\Delta_{cRMSE} = \frac{cRMSE(ISBA) - cRMSE(MEBL)}{\min\left[cRMSE(ISBA), cRMSE(MEBL)\right]} \times 100 \tag{B3}$$

where a positive value corresponds to an improvement using MEBL.





**Table 1.** *Measured litter thickness reported for various sites.*

| Site Référence | Country | Main cover type | Thickness cm | Mass kg m$^{-2}$ |
|---|---|---|---|---|
| Le Bray Ogée and Brunet (2002) | France | Maritime Pine nedle | 5 constant | 2.6 |
| Oak Ridge Wilson et al. (2012) | USA | deciduous oak leaf | 4 constant | 0.6 |
| Kyushu Univ. Sato et al. (2004) | Japan | C.japonica nedle | 5.2 March 2002 | 1.7 |
| Kyushu Univ. Sato et al. (2004) | Japan | L.edulis leaf | 8.6 March 2002 | 2.1 |
| Schaap and Bouten (1997) | Netherland | Douglas fir nedle | 5 | |
| Vorobeichik (1997) | Russia | Taiga leaf | 4.2 July 1990 | |
| Vorobeichik (1997) | Russia | Taiga nedle | 5.4 July 1990 | |
| CSIR Bulcock and Jewitt (2012) | South Africa | Eucalyptus grandis leaves | 3.8 march 2011 | 2.3 |
| CSIR Bulcock and Jewitt (2012) | South Africa | Acacia mearnsii nedle | 1.8 march 2011 | 2.4 |
| CSIR Bulcock and Jewitt (2012) | South Africa | Pinus patula nedle | 4.5 march 2011 | 3.3 |
| Tumbarumba Haverd and Cuntz (2010) | Australia | Eucalyptus leaves | 3 | |
| Wu et al. (2013) | Canada | White pine needle | 2.5 | |
| Marsh and Pearman (1997) | Ecuador | Ocotea infrafoveolata leaf | 1.8 (average) | |
| Mawphlang Arunachalam and Arunachalam (2000) | India | Quercus griffithii leaf | 1.2 (average) | |
| Aoyama Zhu et al. (2003) | Japan | Pinus thunbergii nedle | 4.5 | |
| Seirseminen Haila et al. (1999) | Finland | Spruce nedle | 3-5 | |



**Table 2.** *Characteristics of the three French forest sites.*

| Site | Barbeau | Puéchabon | Bray |
|---|---|---|---|
| Year | 2013 | 2006 | 2006 |
| Localisation | Paris | Montpellier | Bordeaux |
| Main vegetation type | Sessile oak | Green oak | Maritime pine / Grass |
| Climate | Temperate | Mediterranean | Maritime |
| Forest type | Deciduous broadleaf | Evergreen broadleaf | Evergreen needleaf |
| Mean temp (°C) | 10.7 | 13.5 | 12.9 |
| Rainfall (mm) | 680 | 872 | 997 |

**Table 3.** *The main model parameters for each site. Literature indicates that figures come from studies cited in the text and estimated means that values were provided by the principal investigators of each site.*

| Site | Barbeau | Puechabon | Bray | Source |
|---|---|---|---|---|
| Soil Parameters | | | | |
| Sand (%) | 41 | 14 | 98 | literature |
| Clay (%) | 39 | 40 | 2 | literature |
| $W_{sat}$ | 0.36 - 0.48 | 0.06 | 0.42 - 0.42 | estimated |
| $W_{fc}$ | 0.15 - 0.35 | 0.046 | 0.17 - 0.16 | estimated |
| $W_{wilt}$ | 0.05 | 0.018 | 0.04 - 0.03 | estimated |
| SOC 0-30 cm (kg m$^{-2}$) | 5 | 5 | 4.8 | HWSD |
| SOC 70-100 cm (kg.m$^{-2}$) | 4 | 5.5 | 9 | HWSD |
| Root Depth (m) | 1.2 | 5.0 | 0.8 | literature |
| Soil albedo (vis / nir) | 0.03 / 0.13 | 0.05 / 0.2 | 0.1 / 0.18 | estimated |
| Vegetation Parameters | | | | |
| LAI (m$^2$ m$^{-2}$) | 0.5 - 6.0 | 2.4 | 2.0-4.0 | literature |
| Height (m) | 27 | 5.5 | 18 | literature |
| Vegetation albedo (vis / nir) | 0.05 / 0.20 | 0.03 / 0.17 | 0.04 / 0.17 | estimated |
| Vegetation fraction | 0.95 | 0.99 | 0.95 | ECOCLIMAP |
| Litter Parameters | | | | |
| Thickness (m) | 0.03 | 0.01 | 0.05 | estimated |





**Table 4.** The root mean square error, RMSE, square correlation coefficient, $R^2$, and annual error, AE, for the three French sites for the three different experiments.

| Site | Flux | MEBL | | | MEB | | | ISBA | | |
|------|------|------|------|------|------|------|------|------|------|------|
| | | RMSE | $R^2$ | AE | RMSE | $R^2$ | AE | RMSE | $R^2$ | AE |
| Bray | $SW_{net}$ | 3.0 | 1.0 | -0.1 | 3.0 | 1.0 | -0.1 | 3.1 | 1.0 | 0.5 |
| | $LW_{net}$ | 7.1 | 0.95 | 2.3 | 6.0 | 0.97 | 2.1 | 7.1 | 0.95 | -0.2 |
| | $H$ | 52.4 | 0.84 | 5.7 | 52.5 | 0.83 | 4.7 | 63.1 | 0.77 | -2.1 |
| | $LE$ | 59.2 | 0.54 | -1.2 | 60.8 | 0.5 | -1.5 | 58.9 | 0.55 | 0.8 |
| | $G$ | 9.6 | 0.75 | -3.3 | 21.6 | 0.55 | -3.4 | 42.1 | 0.36 | -3.0 |
| Puechabon | $SW_{net}$ | 6.4 | 0.97 | -1.3 | 6.4 | 0.97 | -1.3 | 6.4 | 0.97 | -2.5 |
| | $LW_{net}$ | 7.2 | 0.98 | -0.9 | 5.8 | 0.99 | -1.3 | 9.3 | 0.98 | 4.2 |
| | $H$ | 43.7 | 0.95 | 4.1 | 46.4 | 0.94 | 2.7 | 72.3 | 0.87 | -2.2 |
| | $LE$ | 37.3 | 0.66 | -5.5 | 38.6 | 0.62 | -4.1 | 41.3 | 0.58 | -7.3 |
| | $G$ | 15.8 | 0.81 | 0.8 | 29.2 | 0.78 | 0.5 | 56.9 | 0.48 | 1.4 |
| Barbeau | $SW_{net}$ | 8.7 | 0.96 | -5.9 | 8.7 | 0.96 | -5.9 | 9.4 | 0.97 | -7.6 |
| | $LW_{net}$ | 9.8 | 0.98 | -8.8 | 10.0 | 0.99 | -9.7 | 8.7 | 0.99 | -7.2 |
| | $H$ | 54.1 | 0.8 | 23.1 | 63.8 | 0.67 | 15.9 | 51.1 | 0.8 | 17.2 |
| | $LE$ | 53.4 | 0.76 | -15.1 | 59.6 | 0.66 | -7.3 | 48.6 | 0.78 | -10.2 |
| | $G$ | 4.7 | 0.75 | -3.2 | 22.0 | 0.55 | -3.3 | 42.4 | 0.36 | -2.8 |





**Table 5.** The root mean square error, RMSE, square correlation coefficient, $R^2$ and annual error, AE, computed with available soil temperatures of each site for the three different experiments

| Site | Depth | MEBL | | | MEB | | | ISBA | | |
|------|-------|------|------|------|------|------|------|------|------|------|
| | (m) | RMSE | $R^2$ | AE | RMSE | $R^2$ | AE | RMSE | $R^2$ | AE |
| Bray | 0.04 | 1.9 | 0.98 | -1.0 | 2.9 | 0.96 | -1.3 | 3.6 | 0.93 | 0.1 |
| | 0.32 | 2.3 | 0.97 | -0.9 | 2.8 | 0.97 | -1.2 | 3.0 | 0.96 | 0.2 |
| | 1.0 | 3.0 | 0.94 | -1.1 | 3.5 | 0.91 | -1.2 | 3.8 | 0.89 | 0.2 |
| Puechabon | 0.10 | 3.3 | 0.91 | -0.9 | 3.8 | 0.92 | -1.2 | 4.2 | 0.89 | 0.9 |
| Barbeau | 0.04 | 1.4 | 0.98 | -0.4 | 2.8 | 0.93 | -0.9 | 3.9 | 0.89 | 0.5 |
| | 0.08 | 1.4 | 0.98 | -0.4 | 2.7 | 0.93 | -0.9 | 3.7 | 0.91 | 0.5 |
| | 0.16 | 1.4 | 0.97 | -0.5 | 2.5 | 0.94 | -0.9 | 3.2 | 0.92 | 0.4 |
| | 0.32 | 1.3 | 0.98 | -0.6 | 2.2 | 0.95 | -1.0 | 2.7 | 0.94 | 0.3 |





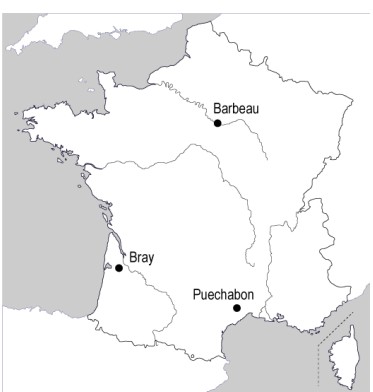

**Figure 1.** The location of the three well-instrumented forested sites in France.

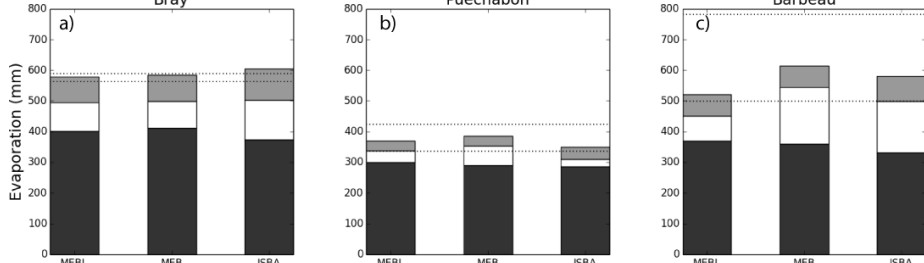

**Figure 2.** The partitioning of latent heat flux for each site and model option into: transpiration (black), ground/litter evaporation (white) and evaporation from the canopy (grey). Values for Le Bray, Puechabon and Barbeau are shown in panels a, b and c, respectively.





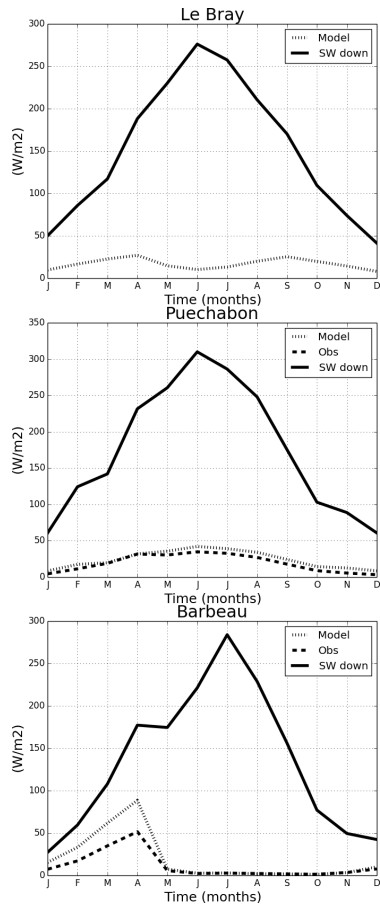

**Figure 3.** The modelled (thin dashed line) and observed (thick dashed line) incoming short wave radiation transmitted through the canopy at Puéchabon and Barbeau . The observations are based on below-canopy $PAR$ measurements. The total incoming short wave radiation is also plotted (full black line) as a reference.





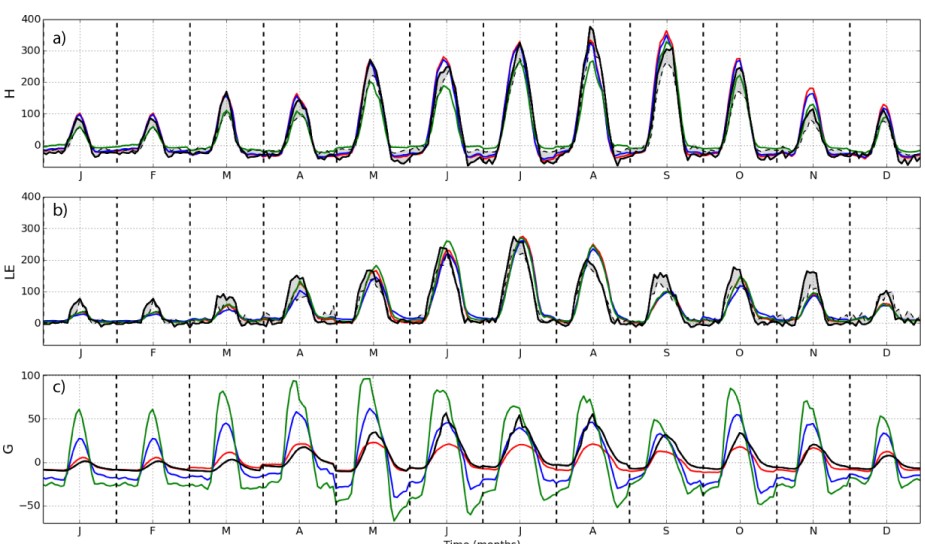

**Figure 4.** The monthly diurnal cycle composite at Le Bray. MEBL is in red, MEB in blue, ISBA in green, measurements are indciated by a dashed black line and adjusted measurements are represented using a solid black curve. As a visual aid, the area between the latter two curves is shaded. Ideally, model results fall within this area.

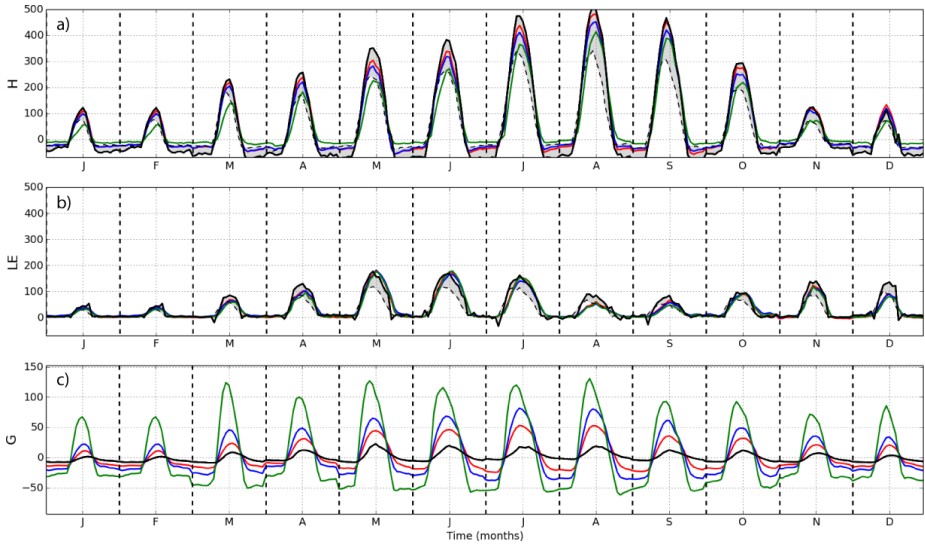

**Figure 5.** As in Fig.4 except for Puechabon.





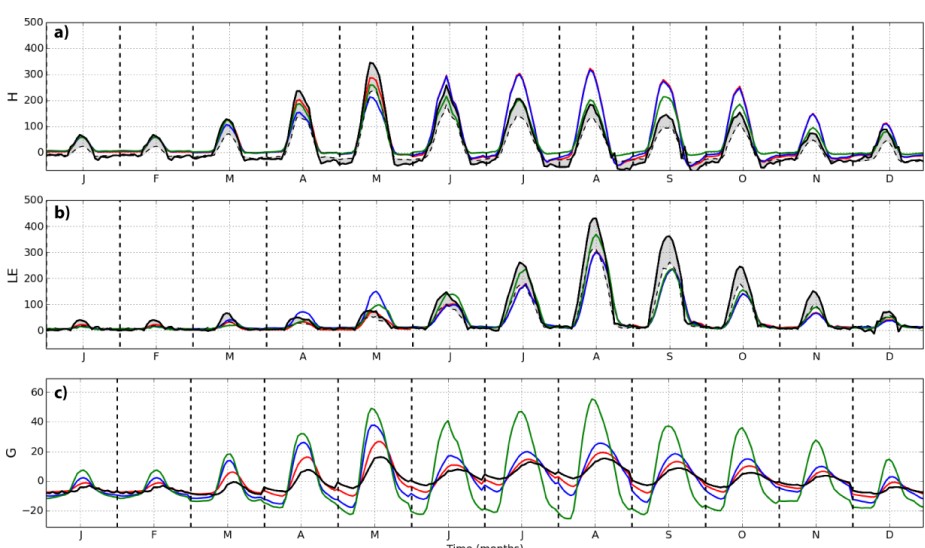

**Figure 6.** As in Fig.4 except for Barbeau.

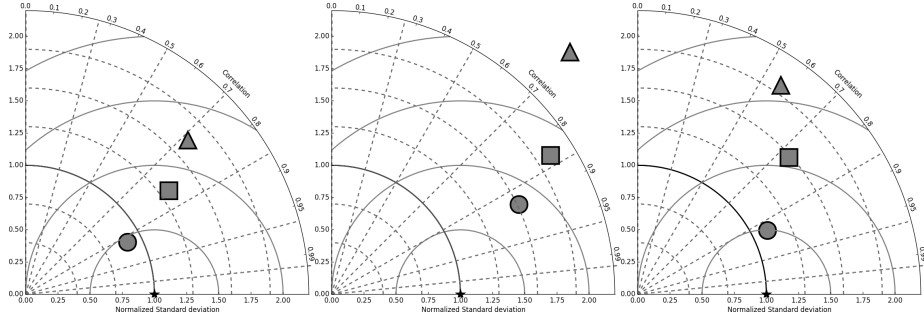

**Figure 7.** The Taylor diagram for ground heat flux, $G$, at Le Bray (panel a), Puechabon (b), and Barbeau (c). MEBL is represented by a circle, MEB by a square, ISBA by a triangle and measurements are indicated using a star. Three scores are represented in each diagram; the correlation coefficient (R) is the angle of the polar plot, the normalized standard deviation (NSDV) is the radial distance from the origin, and the normalized centered root mean square error (cRMSE) is proportional to the distance with the reference (star).





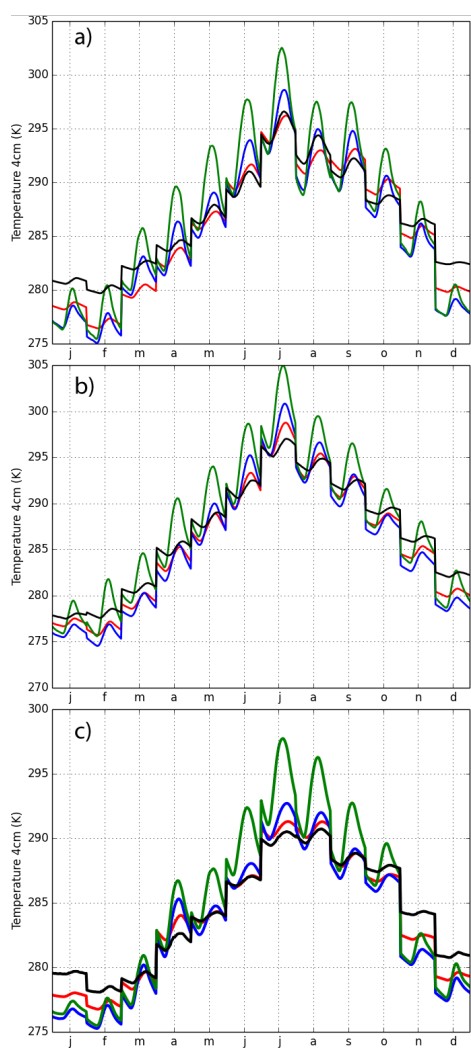

**Figure 8.** Monthly average soil temperature (K) diurnal cycle (at 0.04 m soil depth) composites at Le Bray (panel a), Puechabon (b) and Barbeau (c). MEBL is in red, MEB in blue, ISBA in green and the observations are shown in black.





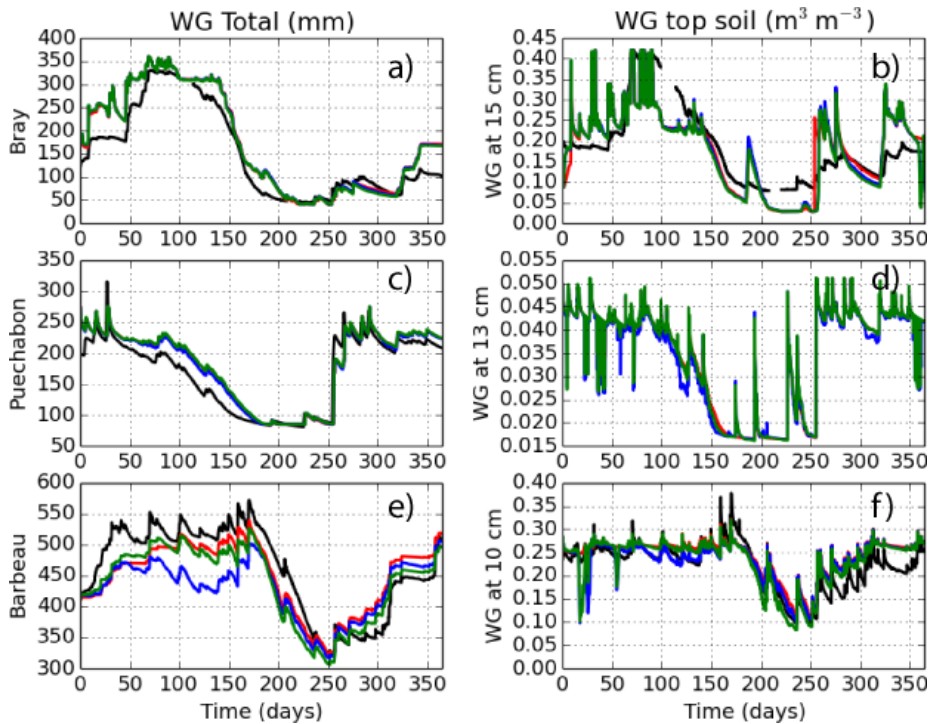

**Figure 9.** The total soil water content (left: panels a, c and e) and the near-surface volumetric soil water content (right: panels b, d and f) at each site. Observations are in black (for Puechabon, the black curve corresponds to the output from a site-specific calibrated reference model from Lempereur et al. (2015), see text). Results for MEBL are in red, for MEB in blue and for ISBA in green.

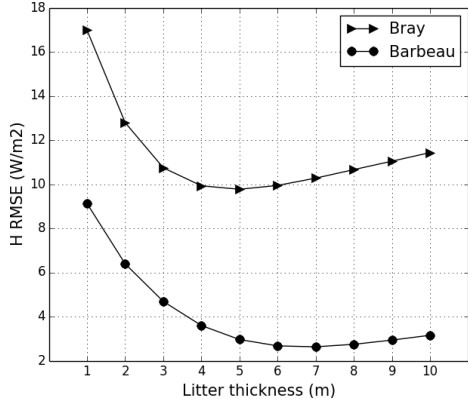

**Figure 10.** The *G* RMSE computed at Le Bray and Barbeau for different values of the litter thickness.

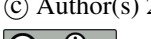



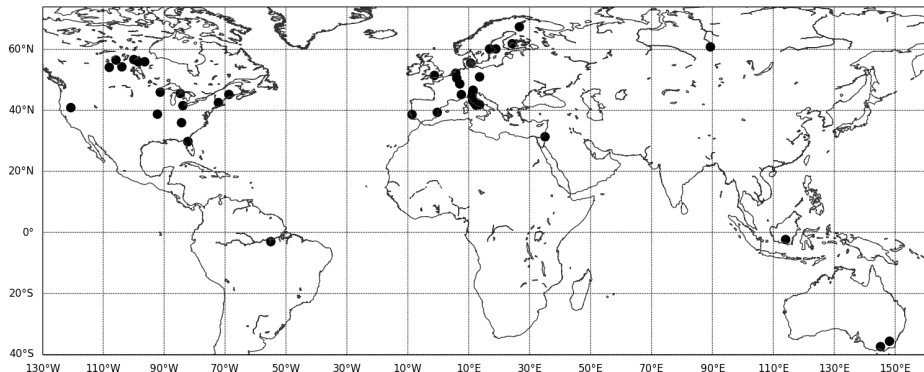

**Figure 11.** The forested sites from the Fluxnet network. Selected sites (shown) have a maximum energy imbalance at or below 20%. The circles indicate the location of sites retained for this study.

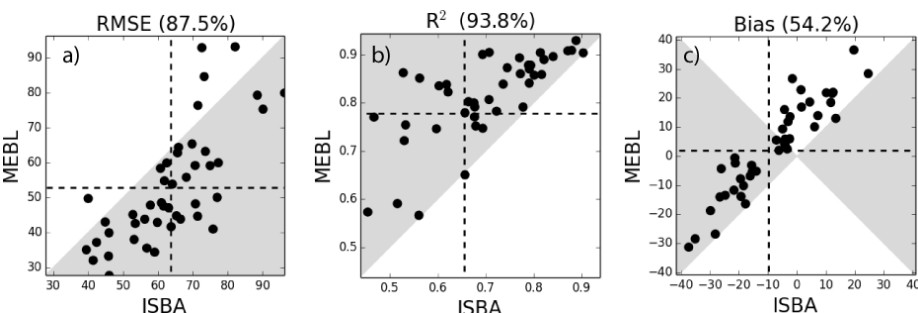

**Figure 12.** Scatter plots of RMSE, $R^2$ and AE doe sensible heat flux ($H$) for each site and year. The abscissa corresponds to the ISBA model and the ordinate to the MEB-L model. Points falling within the grey shaded area imply a better statistical score for MEBL. Each dot corresponds to one site. The intersection of the dashed lines corresponds to the average over all 42 sites. The percentages at the top of each panel correspond to the percent of sites which have better statictics for MEBL.

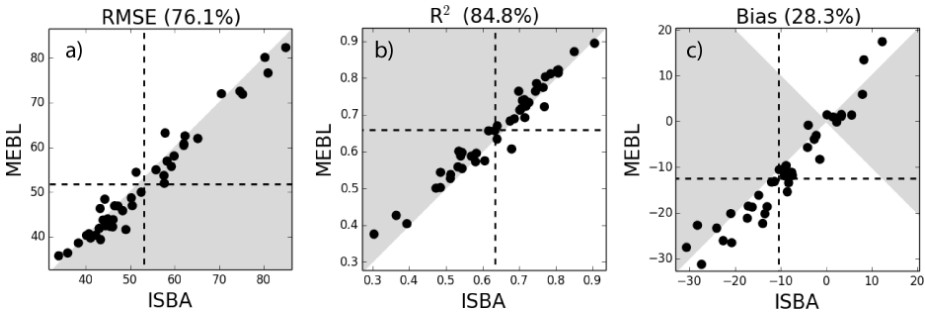

**Figure 13.** As in Fig.12 except for latent heat flux (LE)





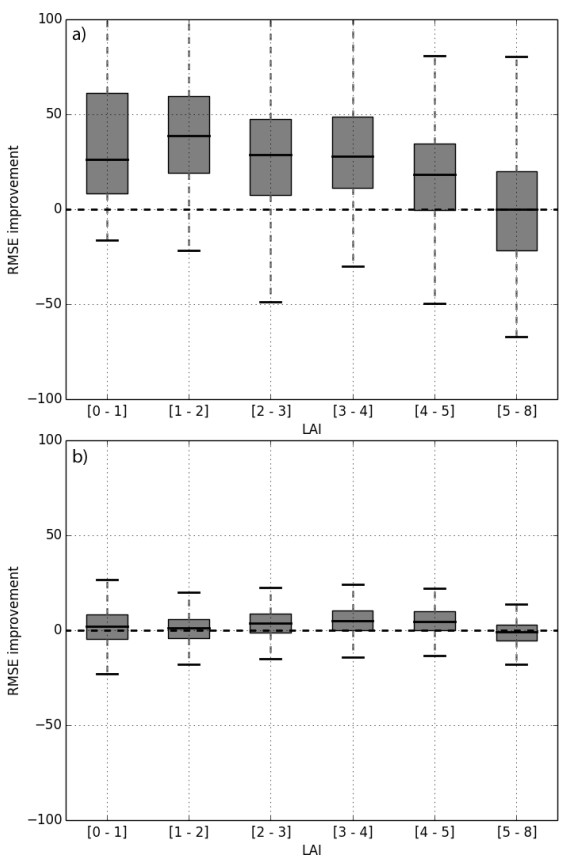

**Figure 14.** Boxplots of improvement in RMSE between MEBL and ISBA. Each RMSE value is computed with over a month-long period which satisfies the energy budget closure condition. The sensible heat flux is shown in panel a, and the latent heat flux is in panel b. The box plots contain 6, 13, 19, 27, 21, and 14 % of all considered months (values) for ranges in LAI increasing from 0 to 1 $m^2$ $m^{-2}$, to 5-8 $m^2$ $m^{-2}$, respectively.