# Peer review of "The Interactions between Soil-Biosphere-Atmosphere (ISBA) land surface model Multi-Energy Balance (MEB) option in SURFEX - Part 2: Model evaluation for local scale forest sites"

_Geoscientific Model Development, 2016_

## Short Comment (SC1) · 30 Nov 2016

Dear authors,

In my role as Executive editor of GMD, I would like to bring to your attention our Editorial version 1.1:

http://www.geosci-model-dev.net/8/3487/2015/gmd-8-3487-2015.html

This highlights some requirements of papers published in GMD, which is also available on the GMD website in the 'Manuscript Types' section:
http://www.geoscientific-model-development.net/submission/manuscript_types.html

In particular, please note that for your paper, the following requirement has not been met in the Discussions paper:

- "The main paper must give the model name and version number (or other unique identifier) in the title."

Please add a version number for ISBA, MEB and SURFEX in the title upon your revised submission to GMD.

Yours,

Astrid Kerkweg

---

## Referee Comment (RC1) · Anonymous Referee #1 · 21 Jan 2017

The paper is presented as a companion paper of a more technical one which describe the new multi energy balance approach developed and implemented within the interactions between the soil biosphere atmosphere model (ISBA) as part of the SURFEX platform. This second paper propose an offline evaluation of the new explicit bulk layer developments described as the so-called ISBA-MEB version, against three well-instrumented forest sites which cover a range in climate, soils and vegetation characteristics. Moreover, authors presented an adding complexity in introducing an explicit litter layer, detailed and called ISBA-MEBL version. Evaluation of these two new versions of the model (MEB and MEBL) against standard ISBA version are done by investigating

the model new versions to simulate the sensible, latent and ground heat fluxes of the three selected forest sites. Finally, benchmark was performed against observation from 42 forested sites from the global micro-meteorological network (FluxNet).

Paper is well written and well constructed which facilitates reading. The evaluation results over the three forest sites present improvements by the two new versions (MEB and MEBL), on the simulations of mainly sensible and conduction fluxes. Net radiation and evapotranspiration fluxes remain unchanged between versions. Introduction of litter layer resolution in MEBL improve significantly ground conduction fluxes G compared to MEB which resulted in better modelisation of dynamic and amplitude of the soil temperature and consequently sensible heat fluxes.

Benchmark against the 42 forested sites compare standard version of ISBA and MEBL version. Results clearly show an overall improvement of the fluxes by MEBL version compared to ISBA standard version.

The developments presented, as much the multi energy balance in the bulk canopy layer as the introduction of litter layer are a significant advance for the LSM and SUR-FEX applications community.

I have just a comment on the article which does not call into question the quality of the work: the paper is presented as a companion paper of a description paper describing the new model developments (a more technical one) and is supposed to present evaluation and validation of this new model. However, the paper quickly addresses the introduction and evaluation of MEBL (with litter) in parallel with MEB (without litter) version. MEBL is presented as better than MEB on the sites exploited due to litter improvement. Then MEB is no longer used in the benchmark against the 42 sites. It is a bias by the authors, because the sites are all forest, but as reader, we would like to have results also for MEB or at minima conclusion about MEB quality even if results are not shown. In addition, MEB and MEBL are put at the same level. As I understand, MEBL is an option to activate in MEB and not an option of ISBA. ISBA is compare to

ISBA-MEB and MEB to MEBL in forest context. But it's not clearly introduced in such terms. However, in a way, validating MEBL actually amounts to validating MEB. And it's well explained in final of conclusion that there is a lot of prospective in following evaluation of MEB.

I suggest a modification of title according to this last remark:

The Interactions between Soil-Biosphere-Atmosphere land surface model with Multi-Energy Balance option (ISBA-MEB) in SURFEX – Part 2: Introduction of litter formulation and model evaluation for forest sites.

Therefore, I propose minor revisions before accepting the paper for publication.

P3 l90: "the" in excess

P4 l92: reference when introduce DIF option

P5 l126: "that" in excess

P5 l155: is there a condition in residual term such res>=0 ?

P7 l212: reference when introduce ECOCLIMAP database

P9 l272: 'is' in excess

P9 l274: is it possible to precise "veg" default value for forests

P9 l287: suppose fig. 3c instead of fig. 3b

P11 l359-361: Fig2 don't present very clearly that both MEB simulations simulate less ground evaporation compare to ISBA, even for MEBL. I suggest to moderate affirmation or link comment to another figure or table.

P13 l 431: ad "at different depth" after soil temperature

P16 l523: I suggest justifying here why only MEBL is considered here.

Legend Table 3: change "indicates that figures come from" by "indicates that values

comes from"

Figure 2: precise if it is partitioning for a specific year or mean of many years

Figure 4 and 5: there is no unit on Y-axis

Legend figure 4: "indicated" and not "indciated"

Figure 7: a,b,c indication are missing

Figure 9: For total WG, please precise soil thickness used to calculus

Figure 10: G RMSE in legend, H RMSE in Y-axis legend. Need to be the same. Litter thickness is in 10-2m, not in m.

Figure 14: Precise H and LE in Y-axis title

---

## Referee Comment (RC2) · Anonymous Referee #2 · 3 Feb 2017

This article evaluates the performance of an improved version of ISBA (as a part of SURPLEX platform) which has been introduced in the first part of the two paper set by the authors. The main improvement to ISBA was the implementing multiple energy balance (MEB) approach and inclusion of a litter canopy layer model. The performance of the improved model in simulating energy fluxes is assessed against standard ISBA and observation over 3 forested site in France and also over 42 Fluxnet network all around the world. I think this article is well written and constructed. Evaluation of the models is done well and clearly shows that the modified version of the model specially implementing the litter model improves the heat flux estimates. Therefore, I recommend this

article for publishing with minor changes.

Title: Change the title to: "The Interactions between Soil-Biosphere-Atmosphere land surface model with a Multi-Energy Balance (ISBA-MEB) option in SURFEXv8 - Part 2: Model evaluation for local scale forest sites"

P3. L83: Remove "from kilometer resolution" or change to "at the resolutions of kilometers"

P4. L92: What DIF stands for? Is it ISBA-DF in Boone et al. 2000? Also correct it in P7 L203.

P4. L 93: change to " The prognostic soil temperature is represented by $T_g$ for $N_g$ soil layers"

P4. L 93: Change to " soil volumetric water content and water content equivalent of frozen water"

P4. 104: Are you referring to the part 1 of this article? Boone et al. 2017a?

Correction should be done in reference section and also line 835.

P4. 108: change to "water content equivalent of snow"

P4. 118: change "models" to "simulates'

P4. 125: Change to "litter water content equivalent of ice"

P6 168: Correct T to $T_g$

P6 191 Correction "assess"

P7 210: Ags??

P7 L 12 : Define ECOCLIMAP

P8 230: Add "leaf are index (LAI)"

P9 L 270 : Change Rnet to Rn as it is in eq 1

P9 L 273 : Remove "is"

P9 279: Is these averages based on the results of the all models at all three sites? For which variable? SWnet?

P9 287: correct Fig 3b to Fig 3c and a,b,c to the figures

P9 294: Based on table 4, RMSE is higher than 8 Wm-2 for Barbeau!

Also, is this average AE for all the models and sites? It does not seem to be correct. Check your calculations please. The values of RMSE and AE are not quite similar across the models as it is mentioned in the text.

P10. 304. Define RCA, replace "one" with "1"

P10. 317. Except with ISBA for Le Bray!

P10 327. Do you mean good results for the sensible heat flux?

P 16. Can you also include a figure for Rnet comparison?

P19 548: New paragraph " In terms of prospective"

P19. 549: MEB or MEBL? Define SIM

Section 5.3: In world wide assessment, is it possible to report also the results with MEB? It would be interesting for readers to see how including litter layer will affect the results.

Table 1. nedle? You mean needle?

Table 2: Correction required : "Mean annual temperature" and "annual rainfall "

Change the number of Fig 11 to Fig 2. Change order of Table 1 and Table 2. Try to number figures and tables in the order that they are mentioned in the text.

Figures 4b and 5b: Hard to see the lines. Can you possible change the scale of the y axis ? also add the units in on Y-axis. Shaded are is not shown well in the figures.

Figure 9: what is the total depth of the soil for WG (better to change it to wg ). Can you specify on the figure?

Figure 10: Correct the Y-axis title to G RMSE also litter thickness should be in cm on X-axis title.

---

## Author Comment (AC1) · 3 Mar 2017

Comment : I suggest a modification of title according to this last remark: The Interactions between Soil-Biosphere-Atmosphere land surface model with Multi- Energy Balance option (ISBA-MEB) in SURFEX – Part 2: Introduction of litter formula- tion and model evaluation for forest sites.

Answer : In accordance with both referee suggestions, we have modified the title to : "The Interactions between Soil-Biosphere-Atmosphere (ISBA) land surface model Multi-Energy Balance (MEB) option in SURFEXv8 - Part 2: Introduction of a litter formulation and model evaluation for local scale forest sites"

Comment : P3 l90: "the" in excess

Answer : corrected

Comment : P4 l92: reference when introduce DIF option

Answer : the reference has been added

Comment : P5 l126: "that" in excess

Answer : corrected

Comment : P5 l155: is there a condition in residual term such res>=0 ?

Answer : We did not apply such a condition. We assume (as seems to be the case generally in the literature as far as we can tell) in the correction method that it is H and LE that are underestimated compared to the available energy Rn-G, but indeed, we understand the relevance of the comment. To explore the sensitivity to this condition, we plotted (see fig1.png) the adjusted sensible heat flux just as it appears in Figs.6-8 for the three sites. Blue curves are plotted using the correction method of the paper and green curves are plotted by only computing the corrections when res>=0, otherwise the turbulent heat flux are not adjusted. Fig1.png shows that the differences are quite small between these two methods and mostly occur during nighttime when the fluxes are relatively weak. We checked the statistics and differences are slight and don't change our conclusions.

Comment : P7 l212: reference when introduce ECOCLIMAP database

Answer : The reference has been added

Comment : P9 l272: 'is' in excess

Answer : corrected

Comment : P9 l274: is it possible to precise "veg" default value for forests

**[GMDD](...)**

Interactive
comment

Answer : the parenthesis is replaced by : "which is constant in time for forests and varies between 0.95 and 0.99 as a function of the forest cover type"

Comment : P9 l287: suppose fig. 3c instead of fig. 3B

Answer : Indeed this is an error and has now been corrected.

Comment : P11 l359-361: Fig2 don't present very clearly that both MEB simulations simulate less ground evaporation compare to ISBA, even for MEBL. I suggest to moderate affirmation or link comment to another figure or table.

Answer : Thank you for this comment, there was a little confusion here. We removed the reference to the figure and better explained that this sentence refers to summer time. We replace these lines by "During this period, both MEB simulations simulate considerably less ground evaporation (about 4 % of summer LE) compared to the standard ISBA simulation which simulates 25 % of summer LE as ground evaporation."

Comment : P13 l 431: ad "at different depths" after soil temperature

Answer : added

Comment : P16 l523: I suggest justifying here why only MEBL is considered here.

Answer : We changed this sentence to : For improved clarity, only the MEBL and the ISBA models are compared here, since the previous evaluation showed the consistent improvement of using the litter option when using MEB for forests.

Comment : Legend Table 3: change "indicates that figures come from" by "indicates that values come from"

Answer : changed

Comment : Figure 2: precise if it is partitioning for a specific year or mean of many years

Answer : The years have been added in the legend

Comment : Figure 4 and 5: there is no unit on Y-axis

Answer : this has been done, the figure has been modified accordingly

Comment : Legend figure 4: "indicated" and not "indciated"

Answer : corrected

Comment : Figure 7: a,b,c indication are missing

Answer : this has been added

Comment : Figure 9: For total WG, please precise soil thickness used to calculus

Answer : we modified the first part of the legend into : "The total soil water content calculated over the root depth indicated in Table.2 "

Comment : Figure 10: G RMSE in legend, H RMSE in Y-axis legend. Need to be the same. Litter thickness is in 10-2m, not in m.

Answer : "H" has been replaced by "G" in the text and the units have been corrected

Comment : Figure 14: Precise H and LE in Y-axis title

Answer : corrected
* * *
[Figure]

**Fig. 1.**

---

## Author Comment (AC2) · 3 Mar 2017

Comment : Change the title to: "The Interactions between Soil-Biosphere-Atmosphere land surface model with a Multi-Energy Balance (ISBA-MEB) option in SURFEXv8 - Part 2: Model evaluation for local scale forest sites"

Answer : In accordance with both referee suggestions, we have modified the title to : "The Interactions between Soil-Biosphere-Atmosphere (ISBA) land surface model Multi-Energy Balance (MEB) option in SURFEXv8 - Part 2: Introduction of a litter formulation and model evaluation for local scale forest sites"

Comment : P3. L83: Remove "from kilometer resolution" or change to "at the resolutions of kilometers"

Answer : The text has been modified to : "This is essential since ISBA is used within the SURFEX platform in various configurations at resolutions ranging from several to just under 10 kilometers at the regional scale, such as within the operational mesoscale numerical weather prediction model AROME, (Seity et al., 2011) and the operational istributed hydrological model system SIM, (Habets et al., 2008), to resolutions ranging from tens to hunderds of kilometers in global scale models, such as within the global climate models CNRM-CM5.1 (Voldoire et al., 2013) and CNRM-ESM1 (Séférian et al., 2015)."

Comment : P4. L92: What DIF stands for? Is it ISBA-DF in Boone et al. 2000? Also correct it in P7 L203.

Answer : A reference has been added. DIF stands for the diffusive option from Decharme et al. 2011 but is replaced by DF for more clarity.

Comment : P4. L 93: change to " The prognostic soil temperature is represented by Tg for Ng soil layers"

Answer : This has been changed

Comment : P4. L 93: Change to " soil volumetric water content and water content equivalent of frozen water"

Answer : This has been changed

Comment : P4. 104: Are you referring to the part 1 of this article? Boone et al. 2017a? Correction should be done in reference section and also line 835.

Answer : This is right, date has been corrected

Comment : P4. 108: change to "water content equivalent of snow"

Answer : This has been changed

Comment : P4. 118: change "models" to "simulates'

Answer : This has been changed

Comment : P4. 125: Change to "litter water content equivalent of ice"

Answer : This has been changed

Comment : P6 168: Correct T to Tg

Answer : This has been corrected

Comment : P6 191 Correction "assess"

Answer : This has been corrected

Comment : P7 210: Ags??

Answer : We clarify what the A-gs definition by changing the sentence to "The canopy resistance formulation is based on the A-gs (leaf net assimilation of CO2 - leaf conductance to water vapour) model (Calvet et al. 1998, Givelin et al. 2006) which simulates photosynthesis and its coupling to the stomatal conductance in response to atmospheric CO2"

Comment : P7 L 12 : Define ECOCLIMAP

Answer : The appropriate reference has been added

Comment : P8 230: Add "leaf are index (LAI)"

Answer : It was defined previously on P4.L116, so we opted not to re-define it here

Comment : P9 L 270 : Change Rnet to Rn as it is in eq 1

Answer : This is done

Comment : P9 L 273 : Remove "is"

[Figure]

Answer : corrected

Comment : P9 279: Is these averages based on the results of the all models at all three sites? For which variable? Swnet?

Answer : It is true that this sentence was not really clear, it has been changed to : "Despite these differences, results are very similar for the SWnet calculation, as seen in Table 4. The models perform well with relatively low values of RMSE ( < 10 W m-2) and annual error (AE < 8 W m-2) for all three sites." With these changes, we hope that it is now more clear that this sentence is about SWnet and that it is valid for each model and site and not the average.

Comment : P9 287: correct Fig 3b to Fig 3c and a,b,c to the figures

Answer : corrected

Comment : P9 294: Based on table 4, RMSE is higher than 8 Wm-2 for Barbeau! Also, is this average AE for all the models and sites? It does not seem to be correct. Check your calculations please. The values of RMSE and AE are not quite similar across the models as it is mentioned in the text.

Answer : Indeed, this is a mistake: the maximum value for RMSE and AE are changed to 10 Wm-2. It is true that the values are not similar in terms of relative difference (e.g. 5.8 and 9.3 W m-2) but for such small values, we think it is more pertinent to consider the absolute differences. We clarify this by modifying the text to : "quite similar in terms of absolute errors".

Comment : P10. 304. Define RCA, replace "one" with "1"

Answer : The sentience is modified to : " A ratio of 1 is used for forests within the default version of the original two-source model in the RCA (Samuelsson et al. 2011) dual-energy budget LSM"

Comment : P10. 317. Except with ISBA for Le Bray!

Answer : this has been added. The text is now worked as : "Except with ISBA at Le Bray, the simulations tend to slightly underestimate"

Comment : P10 327. Do you mean good results for the sensible heat flux?

Answer : We mean good results for LE. This has been clarified by changing the sentence to : "For the Mediterranean forest at Puechabon, most of the net radiation is converted into sensible heat flux (Fig.6a) which leads to low values of LE in accordance with observations (fig.6b)"

Comment : P 16. Can you also include a figure for Rnet comparison?

Answer : Please find attached the figure named fig2.png for an Rn comparison. However, we think that this figure does not add much since there are mostly minor differences between the two models. The main information in fig2.png is that the average over all sites is slightly biased and that RMSE is higher, compared to the three local sites (due to more uncertainties in albedo, LAI . . .) : this is already explained in the text. We can add this figure, but since we feel it does not add alot of additional information and since the paper already includes a lot of figures, we can let this reviewer the editor decide.

Comment : P19 548: New paragraph " In terms of prospective"

Answer : A new paragraph has been designated

Comment : P19. 549: MEB or MEBL? Define SIM

Answer : corrected and the definition of SIM has been added "within the SIM chain (SAFRAN-ISBA-MODCOU, Habets et al., 2008)"

Comment : Section 5.3: In world wide assessment, is it possible to report also the results with MEB? It would be interesting for readers to see how including litter layer will affect the results.

Answer : We plot on fig5.png the results for RMSE calculations for H (left panel) and

LE (right panel). It shows that the litter option almost always performed better for both H and LE. We chose not to show this result mainly for 2 reasons: first, because the paper already includes many figures and we have tried to include those that are the most essential, and second, because the first part of the validation already showed the necessity of the litter representation so that the litter option can be considered for forest as the default version. That being said, if this reviewer think that it is necessary, we propose to add this figure (fig5.png) in the appendix. We propose to refer to it in the text (if it is added) by adding the text: " As mentioned in Section 5.1, significant improvement is obtained with MEB compared to ISBA, and even more improvement with MEBL. An example of the improvement in H and LE between MEB and MEBL is shown in Appendix C Fig.16. But because of the consistently best behavior with MEBL verses ISBA compared to MEB, MEBL has become the default option for forests."

Comment : Table 1. nedle? You mean needle?

Answer : yes, this has been corrected

Comment : Table 2: Correction required : "Mean annual temperature" and "annual rainfall "

Answer : corrected

Comment :Change the number of Fig 11 to Fig 2. Change order of Table 1 and Table 2. Try to number figures and tables in the order that they are mentioned in the text.

Answer : These modifications have been done

Comment : Figures 4b and 5b: Hard to see the lines. Can you possible change the scale of the yaxis ? also add the units in on Y-axis. Shaded are is not shown well in the figures.

Answer : We have plotted both turbulent flux components with the same scale so as the reader can easily understand that the energy from the G mostly goes into H. fig3.png and fig4.png shows figures 4 and 5 of the paper, respectively, with changes to the yaxis

scale. We think that the new figure could bias the reader in terms of the relative weigh of H and LE and that it doesn't add much information since the LE fluxes are quite similar between the models.

Comment : Figure 9: what is the total depth of the soil for WG (better to change it to wg ). Can you specify on the figure?

Answer : the total depth is the root depth indicated in Table 3. This information has been added in the legend.

Comment : Figure 10: Correct the Y-axis title to G RMSE also litter thickness should be in cm on X-axis title

Answer : This change has been done

[Figure]

[Figure]

**Fig. 1.**

[Figure]

[Figure]

**Fig. 2.**

[Figure]

**Fig. 3.**

[Figure]

[Figure]

**Fig. 4.**

---

## Editor Decision (ED1)

Dear author,

Thank you very much for your revised manuscript that, from my point of view, addresses most of the referees' comments. However, I would ask you to consider the following remarks and to modify the manuscript accordingly before its publication.

- Regarding referee #1 comment "P5 l155: is there a condition in residual term such res>=0 ", I would ask you to add in the manuscript a justification about why you did not apply that condition, along the lines of your reply (from my point of view, there is no need of explicitely adding fig1.png in the manuscript).
- p.11, L362, please change "Both MEB simulations ..." for "Both MEB and MEBL simulations ..."
- p.13, L434, I think you added "at different depths" at the wrong place or at least not where the referee suggested. The text "and at at different depths" at the beginning of the line should be removed and "at different depths" should be added after "soil temperatures".
- Figure 14: please also add "H" and "LE" in the captions, respectively after "sensible heat flux" and "latent heat flux".
- P. 3, last paragraph, please simplify! "... resolutions ranging from several to just under 10 kilometers at … to resolutions ranging from tens to hundreds of kilometers in global scale models ..." is too cumbersome. Maybe for something like: "... resolutions ranging from several kilometers at … to resolutions of hundreds of kilometers in global scale models ..."?
- Fig. 3 captions: please change "at Puéchabon and Barbeau" for "at a) Le Bray, b) Puéchabon and c) Barbeau" .
- Your modifications to the first paragraph on p.10 following the referee's comment are not at all clear to me. The first two sentences of the paragraph read: "The simulated LWnet, which depends on the explicit contributions from the soil, vegetation and snow in MEB, and the composite soil-vegetation layer and snow in ISBA, was quite similar in terms of absolute errors among the model versions and led to a fairly good comparison with measurements. The annual RMSE is less or equal than 10 W m−2 for each site and run, and the AE is less than 10 W m−2 (Table 4). '
  - First I do not understand what you mean by "absolute errors" ; do you mean "absolute value of RMSE, R2 and AE"? Even so, I do not understand how this can reconcile the fact that, e.g. 5.8 and 9.3 are very different (as you stress in your reply), or e.g. if you compare LWnet values at Bray for MEBL, MEB and ISBA, i.e. 2.3, 2.1 and -0.2 are very different too (even in terms of absolute value). So I really can't understand how you can conclude: " The simulated Lwnet … was quite similar in terms of absolute errors among the model versions ...".
  - Regarding the maximum values, it would be clearer to write "The annual RMSE and AE absolute values are less or equal than 10 W m−2 and 9.7 W m−2 respectively for each site and run (Table 4).
- On p.10, L307, you added a reference for RCA but it is still not defined.
- On p.10, L308, please change "ratio of one" for "ratio of 1" as you did on L306
- Section 5.3, as you propose, please add fig5.png (which I understand is Fig. 4 of your reply to referee #2) in appendix and please add the text of your reply ("As mentionned in Section 5.1 … has become the default option for forests") in the manuscript. This would also address referee #1 comment about having more results for MEB.

Thank you very much for considering these comments for the final version of the manuscript.

With best regards

---

## Author Response (AR2)

Comment: Regarding referee #1 comment "P5 l155: is there a condition in residual term such res>=0 ", I would ask you to add in the manuscript a justification about why you did not apply that condition, along the lines of your reply (from my point of view, there is no need of explicitely adding fig1.png in the manuscript).

Answer: We added on p.5 L157 the sentence: "Note that there is no specific check here that res>0 which is assume in the correction method. But in order to make sure that this assumption did not impact our results, we recomputed the statistics also checking that res>0 and we found the impact to be negligible."

Comment: p.11, L362, please change "Both MEB simulations ..." for "Both MEB and MEBL simulations ..."

Answer: This has been corrected.

Comment: p.13, L434, I think you added "at different depths" at the wrong place or at least not where the referee suggested. The text "and at at different depths" at the beginning of the line should be removed and "at different depths" should be added after "soil temperatures".

Answer: This has been modified.

Comment: Figure 14: please also add "H" and "LE" in the captions, respectively after "sensible heat flux" and "latent heat flux".

Answer: This has been added.

Comment: P. 3, last paragraph, please simplify! "... resolutions ranging from several to just under 10 kilometers at ... to resolutions ranging from tens to hundreds of kilometers in global scale models ..." is too cumbersome. Maybe for something like: "... resolutions ranging from several kilometers at ... to resolutions of hundreds of kilometers in global scale models ..."?

Answer: The paragraph is modified to : This is essential since ISBA is used within the SURFEX platform in various configurations at resolutions ranging from several kilometers at the regional scale, such as within the operational mesoscale numerical weather prediction model AROME, (Seity et al., 2011) and the operational distributed hydrological model system SIM, (Habets et al., 2008), to resolutions of hundreds of kilometers in global scale models, such as within the global climate models CNRM-CM5.1 (Voldoire et al.,2013) and CNRM-ESM1 (Seferian et al., 2015).

Comment: Fig. 3 captions: please change "at Puéchabon and Barbeau" for "at a) Le Bray, b) Puéchabon and c) Barbeau" .

Answer: This has been corrected.

Comment: Your modifications to the first paragraph on p.10 following the referee's comment are not at all clear to me. The first two sentences of the paragraph read: "The simulated LWnet, which depends on the explicit contributions from the soil, vegetation and snow in MEB, and the composite soil-vegetation layer and snow in ISBA, was quite similar in terms of absolute errors among the model versions and led to a fairly good comparison with measurements. The annual RMSE is less or equal than 10 W m−2 for each site and run, and the AE is less than 10 W m−2 (Table 4). '
◦ First I do not understand what you mean by "absolute errors" ; do you mean "absolute value of RMSE, R2 and AE"? Even so, I do not understand how this can reconcile the fact that, e.g. 5.8 and 9.3 are very different (as you stress in your reply), or e.g. if you compare LWnet values at Bray for MEBL, MEB and ISBA, i.e. 2.3, 2.1 and -0.2 are very different too (even in terms of absolute value). So I really can't understand how you can conclude: " The simulated Lwnet ... was quite similar in terms of absolute errors among the model versions ...".

◦ Regarding the maximum values, it would be clearer to write "The annual RMSE and AE absolute values are less or equal than 10 W m−2 and 9.7 W m−2 respectively for each site and run (Table 4).

Answer: Indeed, this is not clear. We replace these sentences by : " The simulated LWnet, depends on the explicit contributions from the soil, vegetation and snow in MEB, and the composite soil-vegetation layer and snow in ISBA. The annual RMSE and AE absolute values are less or equal than 10 W m−2 and 9.7 W m−2 respectively for each site and run (Table 4). Even if differences can be noticed between the simulations, especially at Puechabon, these errors remain relatively small and comparable, at least compared to the errors of the turbulent and conductive heat fluxes (Table 4).

Comment: On p.10, L307, you added a reference for RCA but it is still not defined.

Answer: We changed the sentence to: "… within the default version of the original two-source model in the RCA ( Rossby Centre Regional Atmosphere Model, Samuelsson et al., 2011) dual-energy budget "

Comment: On p.10, L308, please change "ratio of one" for "ratio of 1" as you did on L306

Answer: This has been changed.

Comment: Section 5.3, as you propose, please add fig5.png (which I understand is Fig. 4 of your reply to referee #2) in appendix and please add the text of your reply ("As mentioned in Section 5.1 ... has become the default option for forests") in the manuscript. This would also address referee #1 comment about having more results for MEB.

Answer: We added the figure and the related comment on p.18 L577. This figure should be placed in the new appendix C section : MEBL vs MEB comparison.

[revised manuscript text omitted]